# Neuronal connected burst cascades bridge macroscale adaptive signatures across arousal states

Brandon R. Munn [1,2,3] ✉, Eli J. Müller [1,2,3], Vicente Medel [1,4], Sharon L. Naismith[1,5], Joseph T. Lizier [3,6], Robert D. Sanders[7,8] & James M. Shine [1,2,3]

The human brain displays a rich repertoire of states that emerge from the microscopic interactions of cortical and subcortical neurons. Difficulties inherent within large-scale simultaneous neuronal recording limit our ability to link biophysical processes at the microscale to emergent macroscopic brain states. Here we introduce a microscale biophysical network model of layer-5 pyramidal neurons that display graded coarse-sampled dynamics matching those observed in macroscale electrophysiological recordings from macaques and humans. We invert our model to identify the neuronal spike and burst dynamics that differentiate unconscious, dreaming, and awake arousal states and provide insights into their functional signatures. We further show that neuromodulatory arousal can mediate different modes of neuronal dynamics around a low-dimensional energy landscape, which in turn changes the response of the model to external stimuli. Our results highlight the promise of multiscale modelling to bridge theories of consciousness across spatio-temporal scales.

The human brain is capable of a rich variety of configurations across levels of arousal, ranging from states of unconsciousness (e.g., sleep or anaesthesia), the perplexing state of dreaming (i.e., externally disconnected consciousness), to the complex, ever-changing patterns that characterise the waking state[1–3]. Different arms of the subcortical ascending arousal system act as the controllers of these global state transitions[4–7] that have long been differentially detected in macroscale empirical recordings (such as electroencephalography−EEG and electrocorticography−ECoG)[8,9]. Despite close contact with empirical signatures of arousal transitions[1,6,10], the precise neurobiological mechanisms by which the specific and nonspecific thalamic nuclei and the ascending arousal system[10–14] (Fig. 1a) interact with cortical

neurons to mediate these arousal state transitions remain poorly understood. Recent advances in the study of anaesthesia have facilitated experimental strategies to differentiate changes in consciousness across arousal[2,8]. However, parallel analytic advances are necessary to illuminate the fundamental neuronal mechanisms responsible for each arousal state.

Testing these ideas empirically is inherently challenging. For practical reasons, the transitions between arousal states have predominantly been studied in coarse-sampled macroscale (i.e., brain regions to whole-brain) recordings. For instance, complex, adaptive signatures such as information theoretic measures of transfer entropy (the directed statistical dependence between a source and a target

[1]Brain and Mind Centre, The University of Sydney, Sydney, NSW, Australia. [2]Complex Systems, School of Physics, University of Sydney, Sydney, NSW, Australia. [3]Centre for Complex Systems, The University of Sydney, Sydney, NSW, Australia. [4]Latin American Brain Health Institute (BrainLat), Universidad Adolfo Ibañez, Santiago, Chile. [5]School of Psychology, Faculty of Science & Charles Perkins Centre, The University of Sydney, Sydney, NSW, Australia. [6]School of Computer Science, The University of Sydney, Sydney, NSW, Australia. [7]Department of Anaesthetics & Institute of Academic Surgery, Royal Prince Alfred Hospital, Camperdown, Australia. [8]Central Clinical School & NHMRC Clinical Trials Centre, The University of Sydney, Sydney, NSW, Australia. ✉e-mail: brandon.munn@sydney.edu.au

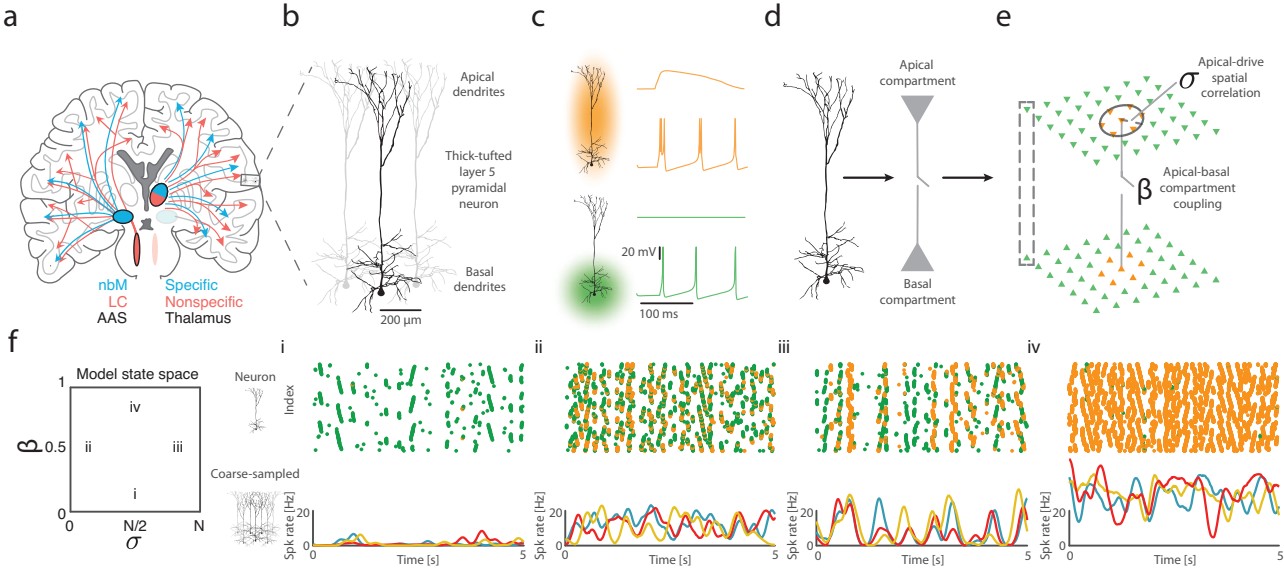

**Fig. 1 | Nonlinear layer 5 pyramidal neurons link across scales and network model architecture. a** The ascending arousal system (AAS) and thalamus are pivotal for controlling arousal state and project to the cortex through specific (targeted) and nonspecific (diffuse) projections, such as the nucleus basalis of Meynert (nbM) and Locus coeruleus (LC). **b** Layer 5 pyramidal neurons dominate macroscale electrophysiology due to their large and parallel dipole moments. Thick-tufted layer 5 pyramidal neurons consist of basal (L4-L6) and apical (L1-L2/3) dendrites that are physically and electrotonically separated by the apical trunk. The two dendritic regions receive differential input across specific/nonspecific projecting thalamic and ascending arousal fibres. **c** Constant current driven into basal dendrites generates regular action potentials ('spikes'; green), whereas simultaneous activation of apical and basal dendrites can transition the cell into high-frequency spiking ('bursts'; yellow). **d** We simulate activity in a network of biophysical dual-compartment pyramidal neurons. **e** We explore two parameters the apical-basal compartment coupling ($\beta$)−controlling the electrotonic threshold required for the apical activity to transition the basal compartment into burst firing −and the spatially correlated apical compartment input ($\sigma$)−whether bursts can occur between adjacent and reciprocally connected neurons, which are modified by subcortical structures. **f** Sub-panels denote different regions of the state space (model parameters) leading to different combinations of (top) neuronal spiking (green) and bursting (yellow), and (bottom) coarse-sampled population activity (normalised per cell) for identical system input−(i) low $\beta$ and intermediate $\sigma$; (ii) intermediate $\beta$ and low $\sigma$; (iii) intermediate $\beta$ and high $\sigma$; and (iv) high $\beta$ and intermediate $\sigma$.

time series[15–18], integrated information (the information integrated by the interactions within the system[19,20]), topological integration (systems-level coordination of brain regions[10,21]), and temporal complexity (the 'dictionary' size required to recreate the signal[22,23]) have all been shown to change across levels of arousal. However, our understanding of the underlying neuronal-level processes that support these emergent properties is limited, due to a difficulty in simultaneous large-scale neuronal recordings and the complexity of multiscale modelling[24,25]. As a result, the neuronal dynamics that facilitate the differential functional properties of a given arousal state (e.g., responsivity and variability to an external stimuli) and their information processing modes, remain elusive.

Although the neuronal features that support arousal are diverse and varied, there is emerging evidence that burst-firing in thick-tufted layer 5 pyramidal neurons (L5$_{PN}$) plays an integral causal role[26–29]. Distinctive among cortical pyramidal neurons, inputs to L5$_{PN}$ basal and apical dendrites (Fig. 1b) are physically and electrotonically separated by a protracted apical trunk. Due to this physical separation, apical dendritic post-synaptic potentials do not typically induce basal somatic action-potentials ('spikes')−as the voltage attenuation along the apical trunk is effectively enhanced by dense hyperpolarisation-activated cyclic nucleotide-gated channels within the apical integration zone[30,31]. However, if the apical input exceeds this electrotonic separation, apical calcium spikes are able to propagate down the apical trunk, and convert coincident somatic sodium regular spikes (Fig. 1c, green) into a high-frequency burst of spikes (Fig. 1c, yellow)[32,33].

Anatomically, L5$_{PN}$ exist at the intersection of various cortico-cortical and subcortical neural streams that are heavily intertwined with the arousal system. For instance, L5$_{PN}$ apical dendrites receive top-down cortical feedback[34], diffusely projecting thalamic projections[35–37], along with neuromodulatory inputs from the

cholinergic (ACh)[38] and noradrenergic (NAd)[39] systems (Fig. 1a). In contrast, the basal dendrites receive bottom-up feedforward cortico-cortical and targeted thalamocortical input[32,36]. Notably, the coupling between apical and basal compartments (and hence the capability to burst-fire) in L5$_{PN}$ has recently been shown to be disabled by a range of anaesthetic agents, suggesting that L5$_{PN}$ are sensitive to changes in arousal[33]. However, advances are required to link these neuronal mechanisms to emergent dynamics at the scale of whole-brain recordings.

Macroscopic recordings of neural activity, such as electro-encephalography (EEG) and electrocorticography (ECoG), are predominantly reflective of summed activity emerging from pyramidal neurons aligned perpendicular to the electrode[24,40]. Of pyramidal neurons, which are the predominant excitatory cells in the cerebral cortex, the largest is the L5$_{PN}$[41]. Due to its size and perpendicular orientation to the pial surface, L5$_{PN}$ are capable of the most significant dipole moment (in contrast to smaller supragranular pyramidal neurons or spherical interneurons) and contribute significantly to macroscale electrophysiological measurements[24]. Thus, L5$_{PN}$ spiking dynamics are dependent upon both the state of the cell as well as the broader network, and their activity is largely responsible for the empirical signatures estimated from macroscale recordings. This leads to a clear, testable prediction: the activity of coordinated bursting in populations of L5$_{PN}$ should recreate key macroscopic signatures of the awake-conscious state. Unfortunately, large-scale recording from L5$_{PN}$ activity across arousal states is technically prohibitive, which impedes our ability to test their cross-scale role directly.

To remedy this problem, we created a biophysically reduced dual-compartment network of L5$_{PN}$ with subcortically mediated cellular properties that we could use to interrogate the impact of arousal across microscopic and macroscopic scales. Altering these properties

led to heterogenous patterns of population-level dynamics—integration, complexity, and integrated information—that were matched with macroscale ECoG recordings of non-human primates as they transition between sleep and wake[9], as well as with human scalp EEG across the conscious states of wake and self-reported dreaming and anaesthesia-induced loss of consciousness[8]. After fitting the model parameter state-space to macroscopic data, we inverted the model to determine the microscopic features that characterise the firing patterns of $L5_{PN}$ across these arousal states. This allowed us to differentiate arousal states of consciousness (anaesthesia/sleep unconsciousness to dreaming and awake consciousness) across two model axes. Strikingly, the awake regime of our model uniquely supported broad cascades of spatiotemporally connected bursting increasing information storage and transfer entropy. Within the awake regime, neuromodulation altered low-dimensional energy landscape topographies, as well as neural variability and responsivity to a broad range of external stimuli. In this way, we demonstrate how multiscale modelling, when matched with empirical data, can reveal both the underlying neuronal dynamics and functional information processing benefits across arousal states and hence provide a robust empirical validation of theories of arousal[1,3].

## Results

### Nonlinear dual-compartment neuronal network model

We constructed a spatially embedded network of nonlinear neurons, in which each simulated neuron consisted of two coupled compartments —a basal and an apical dendritic compartment (Fig. 1d). This allowed us to recapitulate the multi-compartment coupling and burst firing dynamics in $L5_{PN}$[42]. The apical compartment was modelled as a temporal integrator switch. The presence of coincident apical drive that exceeds the apical-somatic electrotonic separation across the preceding 25 ms[34] caused the apical compartment to switch the somatic spiking properties from a regular spiking to a burst firing mode. By utilising Izhikevich quadratic-integrate and fire neurons[42,43], we simulated biologically plausible spike profiles while retaining computational efficiencies (i.e., avoiding multiple channel kinetics across multiple compartments), which allowed us to simulate the systems-level interactions of thousands of nonlinear $L5_{PN}$. The somatic compartments were coupled to one another via a difference-of-Gaussian (i.e., 'Mexican-hat') synaptic coupling—the sum of an excitatory and inhibitory exponential decay, where excitation exceeds inhibition locally and vice versa at larger spatial scales—which captures both biophysical local excitatory and lateral inhibitory effects[44].

To systematically examine the effects of arousal on $L5_{PN}$, we altered two critical parameters of the model: the amount of apical-to-basal dendritic coupling ($\beta$) and the role of spatially correlated apical dendritic drive ($\sigma$; Fig. 1e). $\beta$ controls the apical-somatic electrotonic threshold required for the apical activity to transition the somatic compartment into burst firing. Many biological factors alter $\beta$, for example, NAd (via the locus coeruleus—LC) α2a receptor-mediated closure of HCN channels along the apical shaft[45]; diffusely projecting thalamic activity targeting oblique dendrites[33]; increased apical drive such as following ACh depolarising $M_1$ receptors[38]; or top-down cortical feedback[30]. $\beta$ ranges from decoupled ($\beta = 0$; Fig. 1f$_i$) where $L5_{PN}$ cannot burst, to coupled ($\beta = 1$; Fig. 1f$_{iv}$) where all $L5_{PN}$ spiking activity consists of bursts. In contrast, $\sigma$ captures the spatiotemporal coordination of bursting mediated by differential spatially correlated profiles of drive to the apical dendrites—modelled by convolving the white-noise apical input with a two-dimensional gaussian with S.D. = $\sigma$— ranging across spatiotemporally decorrelated ($\sigma = 1$; Fig. 1f$_{ii}$) to correlated apical dendritic input ($\sigma = N$; Fig. 1f$_{iii}$). Biologically, nonspecific projections (Fig. 1a red) to the apical layers, such as via the nonspecific thalamus[35,37] or ascending arousal system (such as the LC or dorsal raphe)[45] will increase $\sigma$ promoting connected bursting. In contrast, targeted projections (Fig. 1a blue), such as cholinergic inputs from the

nucleus basalis of Meynert (nbM)[46] or specific thalamic projections[35,37], will decrease $\sigma$, leading to a decrease in connected bursting.

We simulated 20 s of neuronal activity for each parameter combination ('state'), with identical white-noise drive (summation of stochastic afferent spikes[47]) to basal and apical (before $\sigma$ apical spatial smoothing) compartments to compare the role of $\beta$ and $\sigma$ in the emergent dynamics. The combination of nonlinear neurons, apical-basal coupling, and differential apical input was sufficient to create substantial heterogeneity in the emergent spiking dynamics of the model (Fig. 1f). We constrained the model such that the mean firing rate ranged between 2 Hz without bursting ($\beta = 0$) and 30 Hz for maximal bursting ($\beta = 1$), which matches known physiological constraints[30,48].

We then calculated coarse-sampled activity by dividing the network into 100 non-overlapping spatial clusters, following a 10×10 grid. While this mesoscale is multiple orders of magnitude less coarse than typical ECoG or EEG recordings, these patterns can in principle be treated as similar to local field potentials, and we hypothesise that changes in the simulated mesoscale dynamics are informative of changes in empirical macroscale dynamics. In this way, a state is described by a precise combination of $\beta$ and $\sigma$, and the nonlinear network model of $L5_{PN}$ can reproduce differential scales of activity— from microscale neuronal spiking (Fig. 1f, green/yellow dots) to coarse-sampled population activity (Fig. 1f, lines).

The combination of nonlinear neurons and spatiotemporally correlated dendritic input was sufficient to create substantial heterogeneity in the model's emergent, coarse-sampled dynamics. For example, low apical-basal coupling spiking activity is sparse and asynchronous (Fig. 1f$_i$, bottom), whereas for intermediate coupling the population activity is highly variable and asynchronously/synchronously coordinated (Fig. 1f$_{ii}$/1f$_{iii}$) with uncorrelated/correlated apical input, respectively. Furthermore, increasing apical-basal coupling ($\beta$) leads to dense population bursting (Fig. 1f$_{iv}$).

### Bridging neuronal to coarse-sampled signatures of complex, adaptive dynamics

We explored how the multiscale neural dynamics differ from neuron to coarse-sampled across the model state space. At the neuronal scale, the model state-space displays an increase in mean firing rate, with apical-basal compartment coupling ($\beta$) consistent with the increase in bursting (Fig. 2a). To further quantify this dynamical heterogeneity in spiking variability dependent upon the state parameters, we calculated two standard empirical neuronal measurements. First, the spike-count Fano-factor ($FF$; variance/mean), which quantifies the mean-normalised firing rate variability and increased with $\beta$ peaking at maximal apical input correlation ($\sigma$) and intermediate-high values of $\beta$ but decreases again as $\beta$ trended towards unity, suggesting that the network became saturated once bursting was too prevalent (Fig. 2b). Second, we explored how apically driven input may alter emergent correlations between neuronal spiking activity. We found that intermediate $\beta$ was associated with an elevated, albeit low mean pairwise spike-count correlation ($r_{SC}$; $\langle r_{SC} \rangle < 0.15$; Fig. 2c), consistent with experimental predictions[47]. These results demonstrate that a simple dual-compartment model with reciprocally connected nonlinear spiking neurons is capable of supporting substantial heterogeneous spiking dynamics.

We were interested in whether our simple model could recapitulate neural dynamics observed across arousal, such as a sleep-to-wake transition. A wide range of empirical epiphenomena has been linked to changes in arousal; however, for brevity, we considered three paradigmatic signatures of complex, adaptive dynamics known to discriminate arousal from anaesthesia at coarser spatial scales. First, the mean Kolmogorov complexity ($KC$)[6,19,22,49], a univariate measure averaged across each coarse-sampled signal reflecting the 'dictionary' size required to recreate the signal where a larger dictionary suggests a

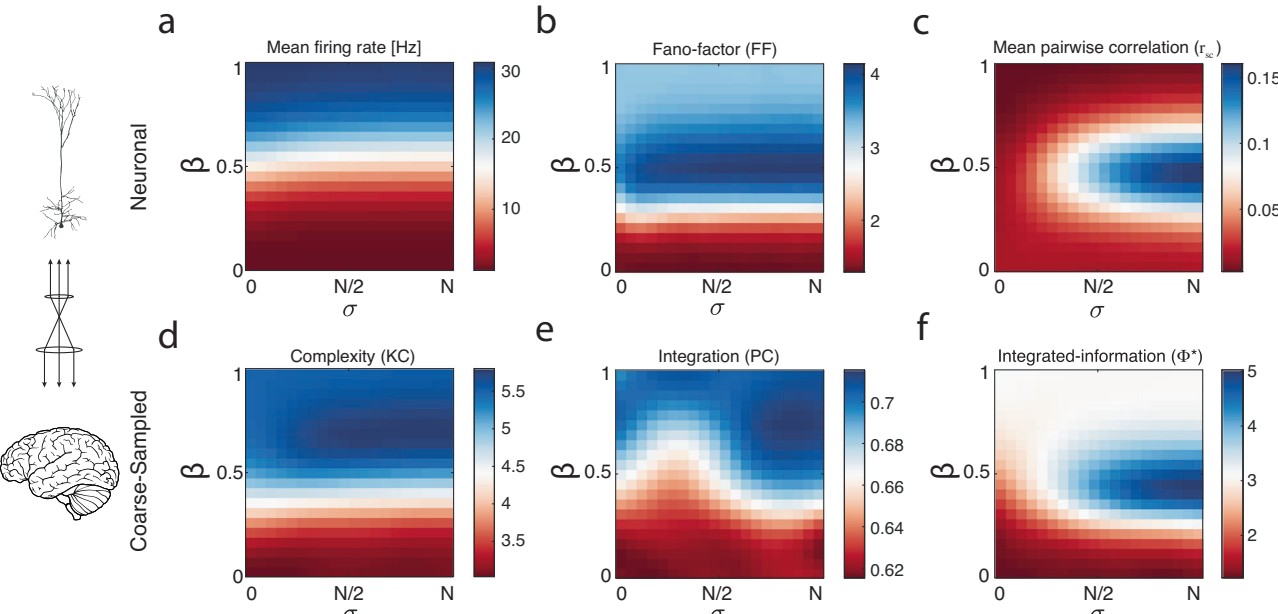

**Fig. 2 | Multiscale modelling allows the comparison between neuronal activity (top) and paradigmatic macroscale signatures of complex, adaptive dynamics (bottom) in the coarse-sampled activity across the model state space. a** mean neuronal firing rate. **b** Mean spike-count Fano-factor (FF; variance/mean). **c** Mean Pearson spike-count pairwise correlation ($\langle\langle r_{SC}\rangle\rangle$). **d** Mean Kolmogorov complexity (KC). **e** Mean participation coefficient (PC). **f** Integrated information ($\Phi^*$).

more 'complex' signal[22,23]. Second, topological integration calculated as the mean participation coefficient (PC)[6], which quantifies the multivariate degree of connection between clusters of coarse-sampled signals (obtained via the Louvain algorithm; see Methods) and has been previously linked to fluctuations in arousal[50]. Last, integrated information ($\Phi^*$) reflecting the multivariate information shared through interactions within the model[19,20].

The three measures varied substantially and distinctly across the model state space. We observed a generalised increase in informational capacity (complexity) proportional to the coupling between apical and somatic dendrites. This increase in information content is consistent with the increased population firing rate[51]. However, we found the maximal complexity occurred with intermediate-high apical-basal dendritic coupling and spatiotemporally correlated apical input (high $\sigma$ high $\beta$; Fig. 2d). Increasing apical-basal coupling increased network integration and correlated apical input (high $\sigma$) L5$_{PN}$ led to a noticeable increase in topological integration that was most pronounced for intermediate $\beta$, where at the two extremes increasing apical-basal coupling led to an increase in network integration (Fig. 2e). The maximal integration aligns with the peak in neuronal Fano-factor and pairwise correlations $\langle r_{SC}\rangle$ (Fig. 2b, c).

We next calculated integrated information ($\Phi^*$; Fig. 2f)[19]. $\Phi^*$ was estimated using mismatched decoding between the coarse-sampled signals and their past at a time-lag of 15 ms chosen as it led to the maximal $\Phi^*$, consistent with empirical findings[9,52]. $\Phi^*$ increased generally with apical-basal coupling, consistent with information complexity (KC) and integration (PC). However, we found that this measure peaked with an admixture of regular spiking and bursting aligning with the peak in neuronal pairwise correlations (Fig. 2c). This suggests that the mixture of both regular spikes ($\beta$ = 50% apical-basal coupling) and spatiotemporally coordinated input ($\sigma \to N$) of L5$_{PN}$ leads to an increase in the integrated information beyond that of the increasing information capacity facilitated generally by increased asynchronous bursting. The coarse-sampled dynamics were obtained using a perfectly non-overlapping subsampling (see Methods); however, the measurements are consistent using partially overlapping coarse-sampling, as would be observed empirically[53,54] (Fig. S1).

Importantly, these three complex, adaptive signatures were selected due to their empirical use and to ensure a diverse set of analytical approaches: for example, KC and $\Phi^*$ reflect univariate and multivariate information theoretic measures, while PC is a topological measure. Their differentiation across the model parameter space emphasises their utility to distinguish differential coarse-sampled dynamics. Another typically utilised signature of macroscale arousal is spectral band-limited power; however, this was not a focus of the study, due to the extensive existing mean-field theoretical studies and empirical fitting linking empirical spectra to thalamocortical resonances (see refs. 55–57). For completeness, we calculated the spectral slope (Fig. S2a)—an indicator of arousal—across the model state-space and found a flattening of the spectral slope with increasing $\beta$ coupling between apical and basal dendritic compartments. That is to say: coupling mediated bursting increases high-power and flattens the spectral slope in a way that was strongly correlated with signal complexity (KC).

## Mapping model parameters to empirical arousal states

To investigate how these different brain-state regimes relate to the empirically observed activity across arousal states, we analysed ECoG recordings from macaque monkeys (*Macaca fuscata*; n = 2) as they transitioned from natural sleep to awake. We segmented the recordings into non-overlapping 20 s epochs to ensure consistency with the model simulations and then calculated the three signatures of complex, adaptive dynamics. In both monkeys, the neural dynamics increased in complexity, integration, and integrated information with arousal (Fig. 3a). The changes in individual signatures across all stages are statistically significant ($p < 0.05$, Kruskal−Wallis with correction for multiple comparisons—see Supplementary Table 1 for a detailed summary).

We next asked whether the different arousal dynamics (i.e., the three-dimensional signatures) would coincide with our model's predicted activity. To test this hypothesis, we utilised a hybrid particle swarm and convex optimisation approach to minimise the summation of the absolute relative difference between empirical and simulated complex, adaptive dynamics across each epoch (see Methods). That is to say, we found the optimal mapping from the three-dimensional

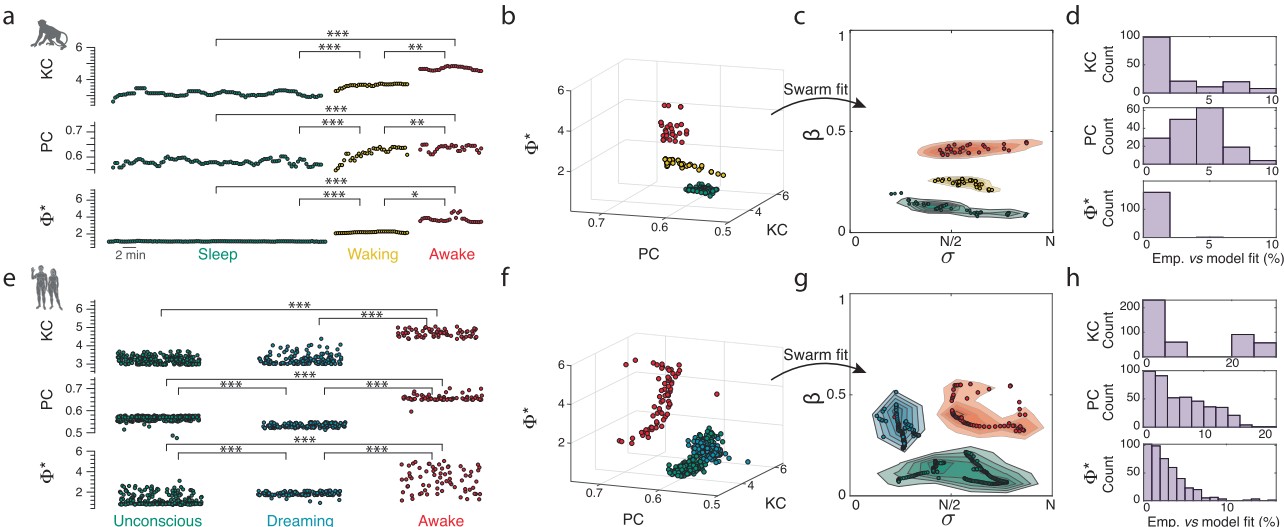

**Fig. 3 | Complex, adaptive dynamics change across arousal states and map to distinct regions of the model state space. a** In macaque EGoG recordings, the three signatures significantly change across consecutive 20 s epoch states of arousal from sleeping (teal) to awake (red). **b** Three-dimensional plot of the three signatures across each epoch which is mapped to the model's state space (Fig. 2d–f) using a hybrid swarm minimisation algorithm. **c** Inverted location of each 20 s epoch in the model state space following hybrid particle swarm/convex optimisation minimising the difference between model and empirical complex, adaptive dynamics where the clouds are five evenly spaced contour lines (2% to 98%) of the probability density estimate for each state. **d** Relative percentage error difference between empirical and model-fitted adaptive signatures. **e** Across 20 s recordings of human EEG under the anaesthetic dexmedetomidine, the three signatures significantly change across unconsciousness (teal), self-reported dreaming (blue), and awake arousal states (red). **f** Three-dimensional plot of the three signatures across each 20 s EEG recording. **g** Inversion of human EEG states to model state-space. **h** Same as in (**d**) for EEG recordings. Statistical significance across empirical complex, adaptive dynamics denoted by *$p < 0.05$, **$p < 0.01$, and ***$p < 0.001$ Kruskal–Wallis multiple comparison tests (see Supplementary Table 1 for (**a**) and (**d**) exact $p$ values).

signatures (Fig. 3b) to the two-dimensional model state space (Fig. 3c) minimising the difference between empirical and model-simulated complex, adaptive signatures. To achieve this we utilised a population-based stochastic particle swarm[58] to search the model parameter space before a fine-scale, interior-point nonlinear convex optimisation was deployed to ensure the discovery of a precise global minimum (see Methods).

We found that the model-inverted sleep, waking, and awake parameters were localised and statistically distinct ($p < 0.01$; Fig. 3c). The teal regions in Fig. 3c demonstrate that the empirical sleep dynamics are optimally matched with simulated coarse-sampled dynamics of L5$_{PN}$ with low coupling ($\beta < 0.2$). Waking moved the matched dynamics through a regime of increased coupling (Fig. 3c yellow), and the awake regime was dominated by transient spatially correlated bursting (Fig. 3c red). Crucially, sleep and awake are significantly different regions of the model parameter space ($p < 0.001$). The optimisation swarm-fitting resulted in a tight match between the complex, adaptive signatures identified in the model and empirical recordings the model-matched measures were within 10% relative percentage error of empirical values (e.g., $\frac{KC_{fit} - KC_{emp}}{KC_{emp}}$; Fig. 3d). To ensure the robustness of our approach, we explored including the spectral slope as a further fitting parameter. Consistent with the other measures, the spectral slope varied across all three arousal states in the macaque recording (Fig. S2b; $p < 0.05$ KW) and the updated fitting led to a subtle change in inverted values (Fig. S3). Furthermore, removing parameters and repeating the fitting procedure led to differential locations, in particular $\Phi^*$, suggesting the three metrics offer a unique discrimination (Fig. S4).

Figure 3c demonstrates that the awake state complex, adaptive dynamics in non-human primate is unlikely to be explained by dynamics: without bursting ($\beta < 0.2$: $p < 0.001$); with only bursting ($\beta > 0.8$: $p < 0.001$); or with spatially uncorrelated bursting ($\sigma < N/4$: $p < 0.001$). Results are consistent when calculated on a separate monkey with sleep clearly distinguishable from waking and awake

states (Fig. S5). However, the waking and awake states overlap at the later stages suggesting the animal awakens faster in this recording.

Next, we asked whether the change in model dynamics could replicate richer arousal states, such as following pharmacologically induced arousal changes with anaesthesia that result in altered states of consciousness (dreaming and awake) or unconsciousness in humans. To assess this, we analysed human EEG recordings under varying levels of the sedative dexmedetomidine, an α2 adrenergic receptor agonist that inhibits the LC, which then indirectly decreases noradrenergic levels in the thalamus and cerebral cortex[59] (thus impairing inter-compartmental coupling in L5$_{PN}$[39]). Using a unique, serial-awakening experimental paradigm[8] wherein patients were periodically woken up from sedation to assess their state of consciousness —recordings were broadly classified into states of awake, unconscious, or disconnected consciousness (i.e., patients reported conscious experience—dreaming—immediately prior to wake up despite being anaesthetised[2]).

The three complex adaptive dynamic signatures were robustly different across the three states of arousal in humans (contrasting unconsciousness, dreaming, and wake; Fig. 3e, f). Specifically, we found that the transition from unconsciousness to wake in humans mirrored the sleep-to-wake transition in macaque—i.e., significant increases in EEG signal complexity, integration, and integrated-information ($p < 0.001$ KW; Fig. 3e teal and red). Further, all three complex, adaptive signatures were heightened in awake, relative to dreaming ($p < 0.001$ KW; Fig. 3e red and blue). Interestingly, we found that dreaming displayed an intermediate $\Phi^*$ that was significantly different to wake and unconsciousness ($p < 0.001$ KW). However, despite this intermediate $\Phi^*$, we found that signal complexity (a proxy of information content) during dreaming was indistinguishable from unconsciousness and integration was significantly decreased (i.e., segregated) from both unconsciousness and wake ($p < 0.001$ KW). This suggests that $\Phi^*$ is detecting an emergent balance between integration and information and the complexity overlap in the dreaming and

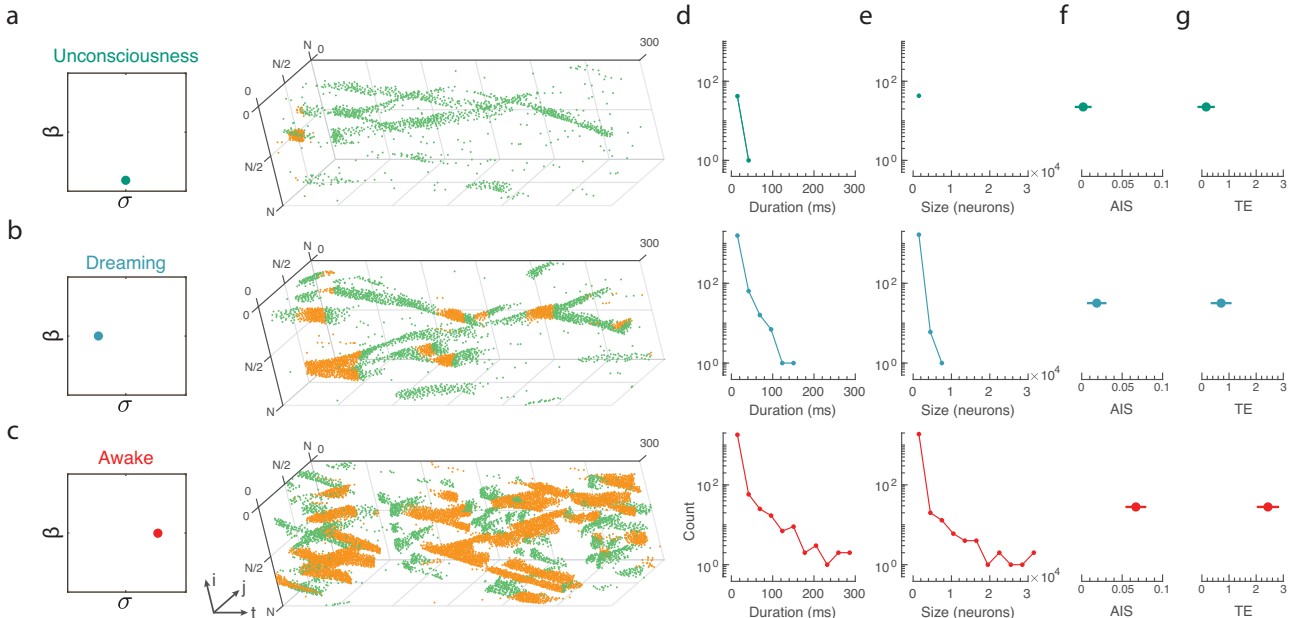

**Fig. 4 | Macroscale awake dynamics state is predicted to be reproduced by microscale spatiotemporally contiguous cascades of bursting neurons.**
**a** Example of the spatiotemporal microscale neuronal regular spiking (green) and bursting (yellow) within the paradigmatic unconsciousness regime of the model (teal dot). **b, c** Same as in (**a**) for (**b**) paradigmatic dreaming (blue dot) and (**c**) awake regimes (red dot). Tracking the connected burst cascades (spatiotemporally contiguous sequences of burst-firing) we quantified the duration (**d**)—the successive timesteps active—and size (**e**)—the number of neuronal bursts, for each of the regimes. **f** Active Information Storage (AIS)—the amount of information a neuron can predict about its future increases with bursting—and (**g**) transfer entropy (TE)—quantifying the information passing between neurons through their bursting activity—across the three regimes. Error-bars are mean ±1.96 SEM (95% CI across 100 random samplings).

unconscious groups may reflect a graded transition in the quality of consciousness (Fig. 3f).

We hypothesised that states of dreaming should explore a different region of the model's state space. Repeating the particle swarm minimisation revealed that the three states of consciousness in humans coincided with three unique regions in the model's brain-state space ($p < 0.001$; Fig. 3g). We found that both the human-anaesthetised and macaque-sleep regimes (Fig. 3g teal) and the human-awake and macaque-awake regimes substantially overlapped in the model state space (Fig. 3g red). The dreaming regime was in an intermediate apical-basal coupled, yet spatiotemporally disconnected, bursting regime ($\sigma < \frac{N}{2}$ and $0.2 < \beta < 0.7$) that was not observed in the primate sleep-to-wake recordings. As dreaming typically occurs around the rapid-eye-movement stage of sleep, which coincides with high ACh levels and low levels of NAd[1], we hypothesise that this is due to cortical ACh increasing L5$_{PN}$ bursting with targeted projections of the nbM (Fig. 1b)[10,60]. The multivariate complex, adaptive signatures of the model state space (i.e., $PC$ & $\Phi^*$) closely matched the empirical measures (all below 15% percentage error; Fig. 3h), whereas the predicted dreaming $KC$ percentage error was ~25% larger than the empirically observed (aligning with unconscious regime), which may reflect either a technical (i.e., EEG coarse-sampling limitation) or model (i.e., non-modelled origin) mismatch.

**Bridging macroscale to microscale across arousal states**
A distinct advantage of our model is that we can bridge across scales and relate macroscale signatures of complex, adaptive dynamics to the underlying microscale neuronal activity. We explored paradigmatic examples of the three diverse regimes of unconsciousness (monkey sleep and human anaesthesia), dreaming (human self-reported), and awake (monkey and human awake) found above. This approach exposed qualitatively distinct patterns of spatiotemporally coordinated burst-firing across arousal states. The unconscious regime activity ($\beta = 0.1$, $\sigma = N/2$) consists of sparse spatiotemporal regular

spikes (Fig. 4a green) with few bursts (Fig. 4a yellow). Figure 4b demonstrates that the dreaming state ($\beta = 0.5$, $\sigma = N/4$; Fig. 4b) consists of an admixture of spikes and bursts. However, the burst spikes are transient and varied. Conversely, the awake state ($\beta = 0.5$, $\sigma = N/4$; Fig. 4c) displays rich spatiotemporal burst sequences ('cascades') propagating across the network. Importantly, it should be emphasised that each of the three simulations receives identical apical and somatic drive for these simulations and that differences in activity are emergent from the apical input spatial correlation ($\sigma$) and apical-somatic coupling ($\beta$).

To quantify this observation, we identified the presence of connected burst cascades, which were defined as spatiotemporally contiguous sequences of burst-firing neurons between pairs of neurons that were causally connected by a structural axon (see Methods). We used the following constraint: for a neuronal burst to belong to an ongoing cascade, the neuron must be physically connected to a bursting neuron in the previous timestep. Results were calculated for temporal windows of 2 ms resolution; however, results are consistent for windows of 0.5 to 5 ms. Across all of the regimes, we calculated the cascade duration (Fig. 4d)—the number of consecutive timesteps a cascade is active—and the cascade size (Fig. 4e)—the number of spatiotemporally connected neuronal bursts across the cascade's duration. Crucially, this analysis is outside the scope of modern empirical neuroscience, as it is currently impossible to simultaneously track burst firing in many simultaneously recorded neurons and know with confidence that any two neurons share a physical synaptic connection.

Tracking these connected burst cascades across arousal states, we found that cascades continued longer in the awake state than during dreaming ($p = 0.025$; Two-sample Kolmogorov–Smirnov goodness-of-fit hypothesis test between awake and dreaming cascades) and unconsciousness ($p = 10^{-20}$), although dreaming cascades lasted substantially longer than those observed during unconsciousness ($p = 10^{-9}$). Interestingly, despite the longer duration, we found that the awake and dreaming cascades possessed a similar distribution of

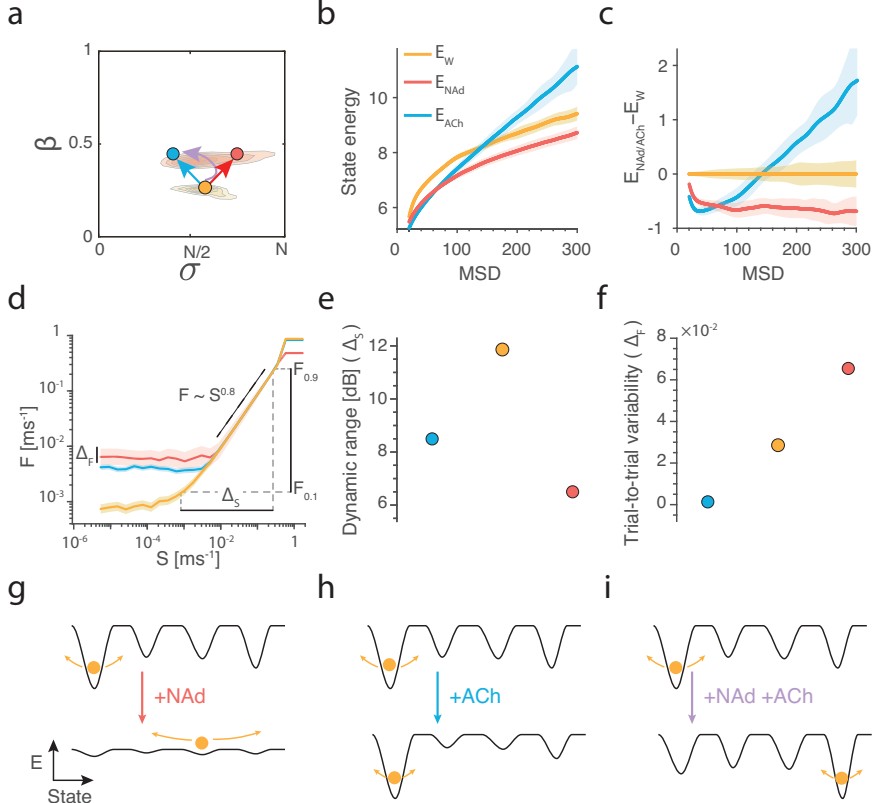

**Fig. 5 | Adrenergic and cholinergic neuromodulation mediate distinct functional information processing modes that modify the energy landscape.**
**a** Comparing the neural dynamics within the awake regime (red Fig. 3c) between low-arousal (yellow), heightened connected bursting such as following noradrenaline (NAd red), and heightened disconnected bursting such as following acetylcholine (ACh blue). **b** Brain-state energy—defined as the mean-squared displacement (MSD) of neuronal firing rates—was calculated across the low-arousal (yellow), NAd (red), and ACh (blue) regimes from (**a**). **c** Relative to baseline, NAd decreases the energy of all brain-state transitions (i.e., makes all state changes more likely). Conversely, ACh increases the energy of large MSD and decreases the energy of small MSD (i.e., quenches variability supporting ongoing dynamics).

**d** Profiles of the population evoked response (F) in the three states across repeated trials to a broad range of input stimuli strength (S). **e** The dynamic range—$\Delta_S$ the discriminable stimuli range—is largest in the low-arousal state. **f** The trial-to-trial variability—$\Delta_F$ the standard deviation in the response across repeated trials of the stimuli—is quenched under cholinergic neuromodulation and enhanced under adrenergic neuromodulation. **g** Conceptually NAd makes it easy to move from one state to another by flattening the landscape (black lines) and increasing state variability (arrows). **h** ACh deepens energy states, meaning the system is tightly locked within the current state with quenched variability. **i** NAd and ACh could synergistically combine to facilitate novel brain-state transitions (NAd) and ensure the transition is realised (ACh). Error-bars are mean ± 95% CI.

cascade sizes (number of spatiotemporal contiguous bursts; $p = 0.06$, where the null hypothesis of similar distributions is just accepted at 95% significance). That is to say: dreaming consisted of dense localised packets of bursting, whereas awake consisted of sparser, more-spatiotemporally connected bursting cascades that travelled across the network. Both the awake and dreaming states possessed larger cascades than unconsciousness (awake vs unconsciousness: $p = 10^{-15}$; dreaming vs unconsciousness: $p = 10^{-8}$). Thus, the awake regime displayed rich interacting long-lasting cascades, whereas dreaming consists of transient, localised large cascades.

These data suggest that connected burst cascades may support the richness of awake consciousness experience. As such, we hypothesised that connected burst cascades should facilitate optimal information processing capabilities. To quantify the information processing capacity of the model, we calculated the active information storage (AIS) and the transfer entropy (TE) of the connected, bursting coalitions. AIS quantifies the amount of information in a sample from a neuronal spiking series that is predictable from its past[17,18]. TE is a directional measure of the conditional mutual information between the past of a laterally connected neuronal spiking time-series process and the present state of a neighbouring neuronal time-series, conditioned on the past of the target (see Methods). Figure 4f demonstrates

that AIS rapidly increased in magnitude from unconscious to awake activity ($p < 0.05$ 95% CI) and from dreaming to awake ($p < 0.05$). Consistently, we found that the TE is the largest in the awake state ($p < 0.05$), with a significant increase in TE between awake and dreaming. That is to say, connected burst cascades in the awake state significantly maximised TE and AIS relative to dreaming and unconsciousness.

## Differential neuronal dynamics within the awake state via the ascending arousal system

At any waking moment, neural activity can rapidly reconfigure due to differential combinations of subcortical drive from the ascending arousal system. Thus far, we have predicted differential neuronal information processing and dynamics across broad arousal states; however, we next wondered how dynamics change at a finer resolution within the waking/awake regime. To do so, we analysed activity between awake and waking (i.e., low arousal; Fig. 5a, yellow) and compared this with larger, more spatiotemporally connected bursting (Fig. 5a, red) and larger, more spatiotemporally disconnected bursting (Fig. 5a, blue). The disconnected bursting state could be reached by targeted anatomical projections such as from the cholinergic nbM (Fig. 5a, blue arrow). Whereas the connected bursting state could be

reached by diffuse projections such as from the adrenergic LC (Fig. 5a, red arrow).

Recent work has begun to interpret neural dynamics through low-dimensional state trajectories[61–63], where the brain state is the instantaneous representation of the underlying neural activity. This trajectory then flows atop the underlying dynamic 'energy' landscape[60]. Through this lens, common changes in activity correspond to 'low-energy' wells in which the trajectory can get trapped, such as an attractor state. In contrast, surprising changes in activity correspond to 'high-energy' states. Here, the term energy follows a statistical physics definition that relates energy to the underlying probability distribution and not the metabolic definition (i.e., the energy used by the brain to maintain or change neural activity). Thus, a statistically energetically expensive state (such as increasing firing rates) may be a metabolically energetically favourable state (and vice-versa). This interpretation is analogous to physical thermodynamics, where low-energy cold particles coalesce, and high-energy hot particles diffuse. Recent fMRI studies have shown that neuromodulators alter the energy landscape of neural BOLD dynamics, both in resting and task-evoked states[60,64,65]. For example, ACh deepens energy wells, and NAd flattens the energy landscape[60]. Based on previous theoretical[10,66] and empirical work[60], we hypothesised that the modulation of the energy landscape detected in BOLD might emerge at the neuronal network scale.

To test the hypothesis that adrenergic and cholinergic neuromodulation differentially alter the topography of the energy landscape, we calculated the energy landscape of the $L5_{PN}$ model across the three regimes. To do this, we first estimated the energy, $E_{\Delta\rho}$, of changes in neuronal firing rate, $\Delta\rho$, defined as $E_{\Delta\rho} = -\ln P_{\Delta\rho}$ where $P_{\Delta\rho}$ is the estimated probability of observing $\Delta\rho$ (see Methods for probability calculation) of the low-arousal state ($E_A$; Fig. 5b, yellow), adrenergic-wake ($E_{NAd}$; Fig. 5b, red), and cholinergic-wake ($E_{ACh}$; Fig. 5b red) regimes. We found that adrenergic neuromodulation flattened the energy landscape relative to the low-arousal state ($E_{NAd} - E_A$; Fig. 5c, red), thus equally facilitating large and small changes in neuronal spiking[67]. In contrast, cholinergic neuromodulation relative to the low-arousal baseline ($E_{ACh} - E_A$; Fig. 5c, blue) stabilised activity and diminished significant changes in firing rate transitions. In this way, neuromodulation differentially modifies the dynamic transitions of neural activity within the awake state.

The analysis thus far has investigated ongoing activity; however, adrenergic and cholinergic neuromodulation has been argued to alter the response profile of neurons to stimuli[12]. Based on these studies, we further hypothesised that cholinergic and adrenergic neuromodulation would differentially augment the network's receptivity to incoming stimuli in a specific way[10]: NAd should increase variability[7], whereas ACh should enhance reliability and selectivity[68]. We calculated the response profiles, F(S), in the three regions of parameter space across a broad range of input (S), and we repeated the stimulation across various trials with different initial conditions (Fig. 5d; see Methods). Finally, we found that the three transfer functions all followed a power-law between their baseline and saturation values with the same scaling exponent $\delta \sim 0.8$, suggesting they efficiently map a large stimuli range to a smaller output, $F(S) \sim S^\delta$, and that the psychophysical Stevens-law is invariant to arousal state (i.e., equivalent differences in stimulus lead to a proportional change in perceived magnitude across arousal[69]).

The shape of the response profile is indicative of the information processing of the neural system. From the response profiles, we calculated the dynamic range, $\Delta_S = 10\log_{10}\left(\frac{S_{0.9}}{S_{0.1}}\right)$, which represents the range of discriminable stimuli[70]. The range $[S_{0.1}, S_{0.9}]$ are inverted from the transfer function $[F_{0.1}, F_{0.9}]$ with $F_x = F_0 + x(F_\infty - F_0)$ where $F_\infty$ and $F_0$ represent the saturation and baseline response, respectively. Another measure we calculated is the trial-to-trial variability of the transfer function, $\Delta_F = \langle \text{Var}(10\log_{10}F(S)) \rangle$, representing the intrinsic

reliability (low $\Delta_F$) or variability (high $\Delta_F$) between a stimulus and output[71]. The low-arousal state possessed the largest dynamic range, $\Delta_S$, (Fig. 5e yellow). Increasing NAd led to the largest trial-to-trial variability, $\Delta_F$, (Fig. 5f red), which is consistent with the theory that NAd facilitates flexible behaviour[67]. In contrast, increasing ACh led to a reduction in variability (Fig. 5f blue), corresponding to an increase in stimuli specificity and reliability, consistent with the known enhancement of stimulus detectability and focus with increased cholinergic tone[68].

The previous two findings within the awake state predict that neuromodulation within the awake state can mediate both neural responsivity and the low-dimensional dynamical manifold. An analogy for the interpretation of these findings is a constantly moving ball in an energy landscape, where the natural movement of the ball corresponds to the neural variability (i.e., black arrows Fig. 5g–i) and the relative depth of a well corresponds to the likelihood of a given brain state (Fig. 5g–i). Following adrenergic neuromodulation, the landscape is flattened—i.e., neural activity can change between previously unattainable states—and the variability of activity is high (Fig. 5g). Conversely, following cholinergic neuromodulation the energy landscape is deepened and neural variability is quenched preventing large and small fluctuations (Fig. 5h). However, isolated neuromodulatory fluctuations are rare. For example, the LC projects to the nbM exciting it via beta2 and alpha1 receptors[12,72,73]. In this case, the synergistic combination of adrenergic-mediated landscape flattening could allow a new brain-state before a rapid cholinergic-mediated deepening ensures the desired brain-state is obtained (Fig. 5i). In this way, the system may be designed to mitigate the excessive and sustained variability following phasic LC activity.

## Multiscale modelling aligns with theoretical sleep modelling regimes

These results confirm prior theoretical work on arousal that mapped conceptual relationships between arousal and brain states, albeit without precise details at the microscopic scale. The empirically constrained regimes of our model state space qualitatively reflect a two-dimensional projection of the three-dimensional Activation (the level of electrical activation in the brain) /Information (the status of gating information flow to and from the brain ranging from endogenously to exogenously driven)/Modulation (the mode of information processing within the brain which is set by the ratio of aminergic to cholinergic modulation) model (AIM; Fig. 6a, cream). In particular, we observe an overlap of the diminished consciousness regime—sleep for macaques (Fig. 6a, teal dark) and anaesthesia for humans (Fig. 6a, teal light)—and two disparate consciousness states, endogenously driven dreaming (Fig. 6a, blue) and exogenously driven wake in humans (Fig. 6a, red light) and macaques (Fig. 6a, red dark). These opposing trajectories of wake vs. dreaming may be explained by the targeted projections from the nbM (cortical source of ACh) and thus decreasing $\sigma$ (vice versa for NAd from the diffusely projecting LC[3,45,46]. Thus, the theoretical manifold introduced by Hobson is qualitatively observed in our dynamics matched empirical brain-state trajectories atop the computational model's parameter space.

As complex, adaptive dynamics change smoothly across the parameter regime, we wondered how many unique states can be distinguished using the three signatures. We explored this question within the model space by clustering the complex adaptive dynamics signatures using $k$ means clustering. Figure 6b (top-left) demonstrates the state space was optimally clustered into $k = 5$ maximally differentiable groups (maximal Davies-Bouldin index for $k = 2$ to 20; see Methods). Thus, despite only using 3 signatures we are able to maximally discriminate 5 macroscale dynamic regimes. How do these regimes map to arousal states observed empirically? At $k = 5$, the clustering solution split the state space into three horizontal bands split for low, low-intermediate, and high apical-

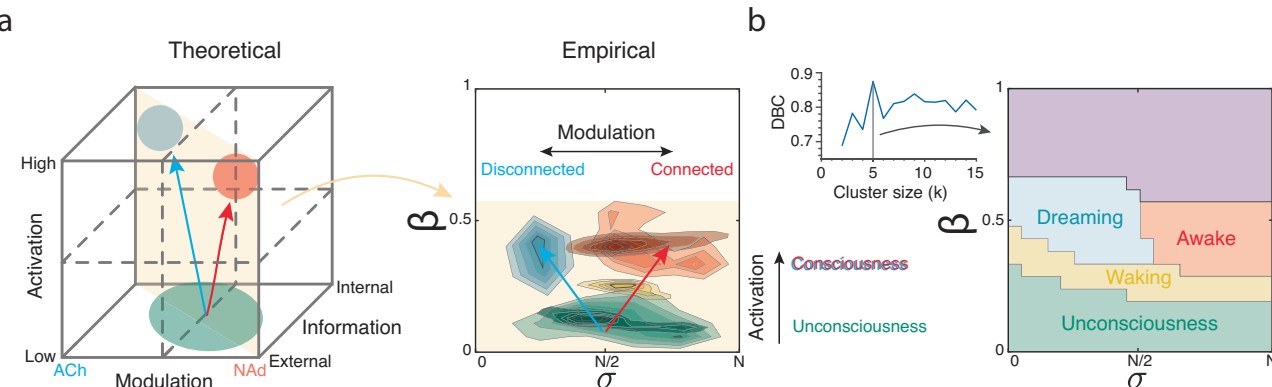

**Fig. 6 | Multiscale modelling and matching across scales confirm theoretical predictions of the Activation/Information/Modulation (AIM) model.**
**a** Theoretical state space of arousal proposed by the AIM model of sleep (reproduced from[1]) where increasing activation moves from a diminished consciousness state to either an awake or dreaming state separated by ratios of cholinergic or monoaminergic neuromodulation. The theoretical state space is closely recapitulated in our empirical coarse-sampled matched model state space, moving from states of unconsciousness (sleep-macaques, dark teal; anaesthesia-humans, light

teal), through intermediate consciousness (waking-yellow), to conscious states separated by disconnected (dreaming-blue) and connected bursting (awake-macaques, dark red; humans, light red). **b** (top-left) The model's complex, adaptive dynamics (Fig. 2d–f) are optimally clustered into 5 clusters as assessed by the Davies-Bouldin criterion (DBC) on 500 stochastic k-means clustering. (right) Four of the five regions overlap with empirically observed states, and a fifth region coinciding with high bursting.

basal coupling $\beta$ and two clusters at intermediate $\beta$ split vertically between disconnected/connected bursting $\sigma$ that differentiates dreaming and wakeful consciousness (Fig. 6b right). The four low $\beta$ state clusters align with the broad empirical states of arousal: unconsciousness (sleep-monkeys/anaesthesia-humans, blue), waking (monkeys, yellow), and awake (humans and monkeys, red), respectively. There is also a clustered state corresponding to extreme bursting which is likely biologically implausible for extended periods due to the maintained high-frequency bursting. Nevertheless, we expect other dynamical signatures would further fine-grain the model's parameter space and improve the fitting procedure for empirical data.

## Discussion

Recent studies emphasise that arousal-consciousness relies on the role of layer 5 pyramidal neurons and their apical-driven capability to switch from modes of regular spiking to bursting[33,74]. How the underlying coordinated activity of large populations of these neurons can give rise to the rich dynamics observed at the systems-scale remains unknown. The combination of multiscale modelling fit to empirical macroscale recordings allowed us to dissect the role of layer 5 pyramidal neuronal bursting across different arousal regimes at a resolution that is currently empirically impossible (Fig. 2/3). We also leverage unique human data that reports reduced arousal in the setting of preserved disconnected consciousness ('dreaming') to model the underlying layer 5 pyramidal neurons changes that subserve changes in arousal states of consciousness. Our model predicts that awake brain dynamics possess cascades of spatiotemporally contiguous bursting in layer 5 pyramidal neurons which are absent in diminished consciousness and limited in 'dreaming' (Fig. 4). These burst cascades permit optimal information processing (Fig. 5), leading us to hypothesise that these burst cascades may reflect a fundamental unit of communication that binds information processing across widespread areas within the cerebral cortex. Finally, this mapping allowed an empirical verification of the theoretical arousal AIM state-space proposed by Hobson (Fig. 6).

We used a percolation-based approach to track burst cascades and investigate whether our model recapitulated these temporal signatures. In contrast to previous approaches[75], we selectively analysed bursting neurons and defined a burst cascade as a sequence of burst-induced burst firing in a set of connected neurons, as it is empirically

difficult to discriminate between regular spikes and bursts using calcium imaging and to know causal anatomical connections. Selecting the active bursts is empirically justified, as consciously perceived events have been linked to coordinated bursts[29], albeit on a narrower spatial scale than employed here. This type of analysis is currently outside the scope of modern in vivo neuroscience, as we do not have access to high-resolution recordings of both activity and structure in the same animals. Nevertheless, our combination of theoretical simulation and matching with multiscale dynamics predicts that connected bursting cascades dominate the underlying neuronal dynamics of the awake state and are absent in diminished states of consciousness and causally disconnected in endogenously driven states of consciousness ('dreaming'). Furthermore, the disconnected bursting cascades may reflect the illogical nonsequential experience of dreaming and the smooth change in complex, adaptive dynamics may represent biological evidence to support the notion that consciousness is graded and changing with information theoretic measures[20].

Connected burst cascades—a theoretically and empirically derived fundamental unit of information processing and communication in the cerebral cortex—represent a parsimonious and unifying framework of Hobson's AIM model of arousal. The mapping to the AIM state space suggests the apical-basal somatic coupling ($\beta$) coincides with the Activation axis which has been empirically linked to EEG activation and cortical firing[76]. In contrast, the spatiotemporally correlated apical input ($\sigma$) coincides with the Modulation axis, where it separates disconnected and connected bursting by the ratio of cholinergic/monoaminergic neuromodulation where we demonstrate an empirically verified correspondence with the theoretical arousal regimes (Figs. 5, 6).

Leveraging multiscale modelling allowed us to build a neuronal model and link its dynamics to macroscale activity through emergent signatures of complex adaptive dynamics. We believe this will be a useful technique for blending empirical and computational work in the future. Nevertheless, the model does not completely predict all dynamics such as the complexity and integration shift during self-reported dreaming and we expect that including additional nuanced components (e.g., NMDA spikes and layer 2/3 dual compartments[77]) will further enhance the discriminability of the model. In addition, our findings suggest that pathological states correspond to precise deformations of burst dynamics. For example, we hypothesise that pathological conditions, such as coma,

narcolepsy, depression, delirium, and epileptic seizures, will be discriminable across the model brain-state space[78,79]. While the critical benefit of this model is in the prediction of microscale neuronal dynamics corresponding to macroscale dynamics, we envisage the concept of matching to global features of complex, adaptive signatures (complexity, integration, and integrated information) will improve the tracking of stages of arousal[63]. In particular, tracking these three signatures would enable a real-time indication of conscious state while under anaesthesia.

Our findings highlight various mechanisms of information processing that can be mediated by varying levels of thalamic or neuromodulatory control. While we found that the awake state maximises transfer entropy and active information storage relative to dreaming and unconsciousness, within the awake state, nuanced changes in arousal can balance opposing information processing requirements. For example, the low-arousal brain is associated with optimal signal detection (sensitivity), which may be evolutionarily beneficial during waking transitions, and the two highly conserved neuromodulatory axes either sharpen specificity and reliability (ACh) or widen the variability of the system (NAd). Finally, the structural connectivity of the locus coeruleus and nucleus basalis of Meynert results in synergistic combinations that can actively overcome adrenergic and cholinergic isolated shortfalls.

In conclusion, we fit a biophysically plausible network model of L5$_{PN}$ to characterise microscopic dynamics across arousal states. In so doing, we demonstrated that the critical feature of the awake, conscious arousal state is the presence of spatiotemporally connected burst cascades of thick-tufted layer 5 pyramidal neurons that alter the information processing of the cerebral cortex.

## Methods

All research complies with the ethical regulations set forth by the University of Sydney, New South Wales, Australia. All subjects provided written consent for each study visit and data were collected in accordance with a protocol approved by the institutional review board at the University of Wisconsin-Madison.

### Layer 5 pyramidal neuronal network model

The layer 5 pyramidal model simulations were obtained via numerical simulation using a phenomenological quadratic integrate-and-fire neuronal model (Izhikevich neurons)[42,43], which is a canonical reduced form of Hodgkin-Huxley neuronal dynamics[80]. The model distils the Hodgkin-Huxley neuronal dynamics to a two-dimensional system of ordinary-differential equations, with 4 dimensionless parameters that can be modified to recapitulate a range of spike-adaptation dynamics that have been observed experimentally[81]. In our simulations we model each neuron with an apical and a basal dendritic compartment. The basal dendritic compartment determines the generation of the spike wave form and dynamics. The apical dendritic compartment serves to shift the spike-adaptation of the somatic dynamics between a mode of regular spikes (no apical intervention) and one of bursting (apical intervention), dependant on the activity within apical dendritic compartment.

### Basal dendritic compartment

First, we define the dynamics of the basal compartment which generates the spikes. The basal somatic dendritic compartment was modelled by the dimensionless membrane equation,

$$\frac{dv}{dt} = h(0.04v^2 + 5v - u + I), \tag{1}$$

$$\frac{du}{dt} = h(a(b(v - v_r) - u)), \tag{2}$$

with the after-spike resetting given by

$$\text{if } v \geq 30, \text{ then } \begin{Bmatrix} v \leftarrow c(t) \\ u \leftarrow u + d(t) \end{Bmatrix}, \tag{3}$$

where the differential equations are in a dimensionless form and these parameters and the constants are reductions to match the spike dynamics to experimentally observed (see[42] for further details) membrane potential, $v$, (mV) and duration $t$ in milliseconds (ms), $v_r$ is the resting potential, and $u$ is the recovery variable, defined as the difference of all inward and outward voltage-gated currents (this emulates the activation (inactivation) of potassium (sodium) ionic currents). $I$ is the input into the somatic dendrites from all sources and $h$ is the integration step, which was set at 0.5 ms. All differential equations in this work were solved using the Euler step numerical integration method, and all analysis was computed on spike-times rounded to the nearest half-millisecond[82–84].

The parameter $a$ represents the time constant of the spike adaptation current and is set as $a = 0.02$. The parameter $b$ describes the sensitivity of the adaptation current to subthreshold fluctuations of the membrane potential ($v$) and is set as $b = 0.2$. The parameters $c$ and $d$ represent the after-spike reset of $v$ and $u$, controlling the voltage reset to model the effect of fast high-threshold K+ conductances and the slow high-threshold Na+ and K+ conductances activated during the spike similarly modulating spike-adaptation as $a$, respectively. The parameters $c$ and $d$ are time-varying and are modified by the apical compartment, as detailed below.

In this paper, we studied a highly recurrent network of L5$_{PN}$, consisting of $N^2 = 70 \times 70 = 4900$ neurons with toroidal grid topology (10 $\mu$m spacing and periodic boundary conditions)[85]. We also simulated dynamics on $50 \times 50$ and $200 \times 200$ grid sizes to ensure spiking dynamics were not affected by finite-size simulations. Afferent connections are made with adjacent neurons falling within a somatic dendritic tree of radius of 200 $\mu$m[85]. Total synaptic currents, $I$, into the somatic dendrites is prescribed by

$$I = I_{ext} + s, \tag{4}$$

where $I_{ext}$ represents the input onto the L5$_{PN}$ from lower-cortical feedforward and subcortical structures. This input was modelled as white noise ($\mu_{I_{ext}} = 0$ mV, $\sigma_{I_{ext}} = 5$ mV) to induce spontaneous activity. For a given neuron $i$, $s_i$ represents the synaptic input from all afferent neurons, while additionally incorporating inhibitory and excitatory neurons. The total synaptic current into a neuron, $i$, is then given by:

$$s_i(t) = \sum_j \sum_k w_{ij} \delta\left(t - t_j^k\right), \tag{5}$$

where $\delta$ is the Kronecker delta function and spike post-synaptic potentials at time $t^k$ from all afferent neurons, $j$, are scaled by a synaptic coupling weight, $w_{ij}$, and summed. The synaptic coupling strength follows a homogenous difference of Gaussians or 'Mexican-hat' function[44] to model the local excitation and lateral inhibition effects[44], given by

$$w_{ij} = \begin{Bmatrix} 0 \text{ if } d_{ij} > d_{max} \text{ or } i = j \\ C_E e^{-\frac{d_{ij}^2}{d_E}} + C_I e^{-\frac{d_{ij}^2}{d_I}} \text{ if } 0 < d_{ij} < d_{max} \end{Bmatrix} \tag{6}$$

where $d_{ij}$ is the Euclidean distance between neuron $i$ and $j$, $C_E$ and $C_I$ are the excitatory and inhibitory coupling constants, and $d_E$ and $d_I$ are the excitatory and inhibitory coupling ranges, respectively. The coupling parameters were set in our model such that the total excitation and inhibition into the network was poised on the border between "dying" and "run-away" activity, which ensures that the excitatory and

inhibitory strengths are consistent with spike rates of spontaneous cortical activity in humans (i.e., ~2 Hz)[86], furthermore they were normalised to the model size to ensure network rescaling preserved approximate spiking dynamics. The coupling parameters utilised in our simulations were $C_E = 180/\sqrt{N}$, $C_I = C_E/2$, $d_E = 1.2\sqrt{N}$, $d_I = 2.5\sqrt{N}$, $d_{max} = 2.5\sqrt{N}$. Finally, the parameters used in the model are set such that the network is balanced, defined as $\sum w_{ij} = 0$, such that the net synaptic coupling into each neuron is zero.

## Apical dendritic compartment
The basal dendritic compartment of each neuron was coupled to its corresponding apical dendritic compartment, whose increased activity could transition the basal dendritic compartment from a regular spiking mode to a burst spiking mode[87–91].

## The spatial correlation of apical input ($\sigma$)
The apical compartment input, $\alpha_i(t)$, into each neuron was generated by convolving independent white-noise drive, $\epsilon_{i,j}$, with a spatial Gaussian kernel $G_\sigma(d_{ij}) = \frac{1}{2\pi\sigma^2}e^{-\frac{|d_{ij}|^2}{2\sigma^2}}$, with spatial decay $\sigma$, which ranges from $\sigma = 0$ (i.e., independent white-noise into each neuron) or $\sigma = N$ (i.e., apical compartment input is strongly spatially correlated). The total apical compartment drive into a neuron, $i$, is then given by

$$\alpha_i(t) = G_\sigma\left(d_{ij}\right) * \epsilon_{i,j}. \tag{7}$$

## Apical-basal electrotonic separation ($\beta$)
The apical input can transition each L5$_{PN}$ from a regular spiking mode to a burst spiking mode, depending on whether the apical input over the previous 25 ms exceeds the electrotonic separation controlled by the coupling between the apical and basal dendritic compartments ($\beta$). This parameter modifies the neuronal spiking variables $c_i(t)$ and $d_i(t)$ following

$$c_i(t) = -65 + 10H\left(\sum_{t'=t-25}^{t}\alpha_i(t') - I_h(\beta)\right), \tag{8}$$

$$d_i(t) = 8 - 4H\left(\sum_{t'=t-25}^{t}\alpha_i(t') - I_h(\beta)\right), \tag{9}$$

with $H$ as the Heaviside step-function and $I_h$ is the electrotonic leak-current, which is a function of the apical-basal coupling $\beta$. This results in two conditions: if the apical activity does not exceed the electrotonic separation, then $c = -65$ and $d = 8$, and the neuron recapitulates regular spiking dynamics, such that when driven with constant input the neuron responds with a short inter-spike interval (ISI) which gradually increases with input amplitude; in contrast, if the apical current exceeds the HCN channel mediated leakage current then, $c = -55$ and $d = 4$, which recapitulates intrinsically bursting spike dynamics, such that when driven with constant input, the neuron responds with bursting, followed by repetitive short ISI spikes[92]. The two regimes of spiking dynamics can be observed in Fig. 1b in the main text. The parameters were chosen based on original research that fit the spike profiles of regular spiking and bursting L5$_{PN}$[42,81]. Thus, a simulated action-potential can be defined as either a burst or a regular spike depending on the $c$ and $d$ parameters at the time of activation.

## Exploring the model state-space according to $\beta$ and $\sigma$
The main impetus of this work was to explore the biophysical phenomenon of coordinated bursting on emergent brain-state dynamics. This orients the exploration of our model to two key parameters corresponding to apical-basal coupling, $\beta$ and reciprocally connected bursting, $\sigma$. In the manuscript, we present results for an HCN channel

mediated leakage current ranging from $I_h(\beta = 0) = -3$ to $I_h(\beta = 1) = 3$ in 20 linear steps, and $\sigma$ ranging from $\sigma = 1$ (disconnected bursting) to $\sigma = N$ (connected bursting) in 42 linear steps. Thus, we ran $40 \times 42 = 1,680$ simulations, with identical temporal drive and apical input, prior to Gaussian convolution. We ran each simulation for 35 s, in time-steps of $\Delta t = 0.5$ ms and discarded the initial 15 s of simulation so as to avoid transient dynamics induced by initial conditions.

## Coarse-sampled neural activity
While the exact relationship between local-field potentials (for electrocorticography (ECoG) and EEG signals) and underlying neural spiking activity is still debated, signals from these modalities represent macroscale summations of all ionic currents[93], including action potentials[94]. Given, L5$_{PN}$ are numerous, large and geometrically aligned, a crucial feature for constructive superposition of extracellular field potentials, and furthermore, they undergo metabolically 'expensive' bursts which are believed to be a primary contributor to these cortical signals[24,95–98]. For these reasons, field potentials are believed to be dominated by L5$_{PN}$ activity[99–102]. We can thus use these relationships to create a mapping between spiking activity and ECoG/EEG. This mapping corresponds with the findings of[103–106], who concluded that the population firing rates are largely responsible for ECoG/EEG signals. Furthermore, for spectral analysis, the data used in our analysis preserves the spiking spectral properties[24,55].

To obtain a spatially coarse population measure from our simulations that is comparable to these empirical recordings, we utilise a standard approach of pooling the neuronal spiking activity[107–109]. We coarse-sample at an intermediate spatial-scale pooling into 100 subpopulations of 49 perfectly nonoverlapping and spatially localised neurons, subsampling across a $10 \times 10$ grid. One alternative approach would be to calculate the average membrane potential[110–114]; however, as the two experimental measures correlate stronger with spiking activity, we opted for the former method. Another alternative approach, aligning with empirical conditions would be to sample spiking activity with overlap across neighbouring regions[53]. To do this, we subsampled spiking activity spatially pooled by convolution with a 2d gaussian with $\sigma = 3$ neurons (i.e., 40 $\mu$m) as if recorded using a $10 \times 10$ multielectrode array. Nevertheless, despite this significantly overlapping coarse-sampling (i.e., significant spurious correlations) the multivariate complex, adaptive signatures are only slightly affected ($PC$ & $\Phi^*$; r > 0.9 perfectly sampled/overlapping sampling) and the univariate ($KC$) is unaffected (Fig. S1).

## Experimental methods
**Statistics and reproducibility.** In this paper, we utilise two datasets for comparison to our model outputs: cortical ECoG data recorded from two macaque monkeys (both male) transitioning from asleep to awake (available from the NeuroTycho public repository; http://www.neurotycho.org) and human EEG data ($n = 20$, 5 female) across varying states of consciousness (unconsciousness, consciousness, and dreaming)[8]. No statistical method was used to predetermine sample size and no data were excluded from the analyses. In the following sections, we describe the justification behind comparing the modelling data and the pre-processing applied to each of these datasets.

**Macaque ECoG recordings.** We analysed freely available ECoG data from the *neurotycho* dataset of macaque monkeys (*Macaca fuscata*) George (Fig. 2a) and Chibi (Fig. S5) and undergoing a sleep experiment. All experimental and surgical procedures were performed in accordance with the experimental protocols approved by the RIKEN ethics committee. In a single recording session, data was captured while the monkeys transitioned from sleep to wake. The full protocol and definition of sleep conditions can be found in ref. 9. Here, we briefly describe the aspects of the protocols that are relevant for our analysis.

The activity was recorded on an array of ECoG electrodes embedded in an insulating silicone sheet. The surface of the sheet was dimpled to expose the surface of ECoG electrodes with a diameter of 1 mm. The electrodes were made of platinum discs, and the inter-electrode distance was 5 mm. The array consisted of 128 electrodes implanted in the subdural space covering the left hemisphere over the frontal, parietal, temporal, and occipital lobes. The experimental and surgical procedure was performed in accordance with the protocols approved by the RIKEN ethics committee. The electrode recorded at a sampling rate of 1 kHz[9].

To remove line noise and reduce artefacts in the ECoG data, we computed bipolar re-referenced signals between two neighbouring electrodes, resulting in a reduction of the 128 electrodes to 64 independent channels. A second-order Butterworth notch filter was applied at 50 Hz to remove line-noise. We analysed the ECoG data recorded from four experiments in which recordings were made continuously while the macaques ($n = 2$) were asleep, waking up, and awake with eyes closed. We divided each of these recordings into non-overlapping epochs of 20 s to ensure consistency with the model simulations.

**Human EEG recordings.** Subjects ($n = 20$, 5 female) were enroled in the UNderstanding Consciousness Connectedness and Intra-Operative Unresponsiveness Study (UN-ConsCIOUS, NCT03284307)[8]. Participants were healthy volunteers between 18 and 40 years old without prior contraindications to anaesthetics. Sex and/or gender was not considered in the study design. All subjects provided written consent for each study visit and data were collected in accordance with a protocol approved by the institutional review board at the University of Wisconsin-Madison. Participants were compensated $200 for taking part in a sedation study. Anaesthesia was administered under the supervision of an anaesthetist to achieve a series of stable drug-dose plateaus. For Dexmedetomidine, a rapid infusion of 3.0 μg kg⁻¹ h⁻¹ was initially administered over a 10 min period followed by a 0.5 μg kg⁻¹ h⁻¹ maintenance infusion to achieve the first drug step. The second step was similarly achieved by a 10 min infusion of 3.0 μg kg⁻¹ h⁻¹ followed by a 1.5 μg kg⁻¹ h⁻¹ maintenance infusion. Subjects were allowed to rest with their eyes closed for 2–10 min at a time. Each rest period was concluded by a researcher calling the participant's name and initiating a brief structured interview consisting of questions designed to assess if the participant had been having a conscious experience directly before the name call and if the experience was connected to the environment through the senses. Participant answers were evaluated by two members of the research team to code each wake report as consciousness, disconnected consciousness (conscious experience but no awareness of the environment i.e., 'dreaming'), or unconsciousness (complete lack of experience)[8,115]. If the subjects were not rousable, they were not presumed unconscious and the attempted wake-up was excluded from the analysis.

High-density EEG data were collected using a NA300 EGI system with 256-channel gel caps. Electrodes were manually prepared with application of electrolyte gel to achieve electrode impedances <50 kΩ. Data were recorded using EGI Net Station Acquisition 5.4 software (Eugene, OR, USA). Data were filtered between 0.1 and 55 Hz. Filtered data were then visually inspected for noisy channels and noisy epochs, which were removed. Independent component analysis was then computed using the InfoMax algorithm, and components dominated by eye movements or muscle artefacts were rejected. After these cleaning steps, data were average referenced, and to avoid volume conduction and reduce point spread, the signal was transformed to current source density by using spatial Laplacian derivatives. Consistent with the model and macaque analysis we analysed the preceding 20 s epochs before the wake report was segmented out for analysis.

## Analytic methods

We now outline the analysis methods applied to the simulated and experimental data.

### Spike statistics

**Inter-spike interval.** Inter-spike intervals (ISI), defined as the time interval between successive spikes in a spike train, were calculated for each neuron. Given $J$ spikes let $t_i$ be the occurrence time of the $i$th spike. The ISI sequence is:

$$\text{ISI} = \{t_2 - t_1, t_3 - t_2, \ldots, t_J - t_{J-1}\}. \tag{10}$$

**Spike count.** To calculate spike-counts, we followed the approach described by[47]. First, time was divided into $dt = 1$ms bins, and a binary spike train, $sp_i(t)$ was created for each neuron, $i$, equal to 1 if there was a spike in $t, (t + dt)$ and 0 otherwise. The spike-count, $n_i(t; T)$ of window size $T$ is defined as the number of spikes in $(t, t + T)$, which can be written as a convolution between the spike train and a square kernel, $K_T$, of length $T$,

$$n_i(t; T) = K_T * sp_i(t) = \sum_{t'} K_T(t' - t) * sp_i(t') \tag{11}$$

where $T = 50$ ms consistent with[47]. We used a box with amplitude $1/T$ in $(t, t + T)$ and zero otherwise to ensure $n_i$ has units of spk/s.

**Fano-factor.** We calculated the mean-normalised spike-count variance or Fano-factor, $FF_i$, for each neuron[116], $i$, calculated as

$$FF_i = \frac{Var(n_i(t))}{\langle n_i(t) \rangle}, \tag{12}$$

where $Var(n_i(t))$ is the variance of the neurons spike-count,

$$Var(n_i(t)) = \sigma^2_{n_i(t)} = \left\langle \left(n_i(t) - \langle n_i(t) \rangle\right)^2 \right\rangle. \tag{13}$$

If spike times are Poisson-distributed, then $FF = 1$ as the variance of a Poisson process is equal to its mean. Any deviation from unity thus implies a divergence from Poisson-like activity, divergence below unity suggests regularity while divergence greater than unity indicates an increase in variability, relative to a Poisson process.

**Spike-count correlation.** The spike-count correlations between neurons $i$ and $j$ were calculated using the standard correlation coefficient, $r_{sc}$, calculated as

$$r_{sc} = \frac{Cov(n_i(t), n_j(t))}{\sqrt{Var(n_i(t)) Var(n_j(t))}}, \tag{14}$$

where $Cov(n_i(t), n_j(t))$ is the covariance between the spike-counts of the two neurons:

$$Cov(n_i(t), n_j(t)) = \left\langle \left(n_i(t) - \langle n_i(t) \rangle\right)\left(n_j(t) - \langle n_j(t) \rangle\right) \right\rangle. \tag{15}$$

**Signatures of complex, adaptive dynamics.** As detailed above, the brain-state signatures were calculated on either the pooled spiking activity of the population, for the Kolmogorov complexity, or between 100 pooled sub-populations equally spaced from the network each consisting of 49 neurons. The pooled spike count, $n(t, T)$, was calculated as above after convolving the population spike vector $dt = 1$ ms with a Gaussian $T = 200$ ms.

**Kolmogorov complexity.** We calculated the complexity of the signal using the Kolmogorov complexity, $KC$, which is a measure of

the information content of a signal. This can be conceptualised as how compressible the signal is without loss of information content. Intuitively, a complex (random) signal is difficult to compress, since sequences are mostly unique, whereas a simple/repeating signal is easily compressible since common phrases in the signal can be encoded in manner cheaper than the complete phrase by assigning them an identifier[22,51]. To compute this measure, the population signal was binarised into 1, if greater than the mean and 0 otherwise. The binary sequence was then scanned from left to right and a complexity counter, $c(m)$, was increased whenever a new sub-sequence of consecutive characters was detected. $KC$ was then calculated as:

$$KC = \frac{c(m)}{m/\log_\alpha m},$$ (16)

where the complexity counter is normalised by the number of 'words', $m$, and the number of 'characters' in the language, $\alpha$, in this case, $\alpha = 2$ a binary language. These measures have been shown to increase with waking/consciousness and decrease with sleeping/anaesthesia[117,118]. The population signal is calculated at the resolution of the relevant signal (i.e., dt = 1 ms model, monkey ECoG, and human EEG).

**Participation Coefficient.** The time series of activity (be that coarse-sampled population spiking activity, bipolar re-referenced ECoG, or surface Laplacian EEG), $n$, were used to create a weighted, signed, and un-thresholded functional connectivity matrix (using the region-to-region Pearson's correlation as a measure of functional connection strength), which we then examined for a modular network[119]. The algorithm optimises a multilayer modularity quality function, $Q$, using a weighted- and signed- version of the Louvain modularity algorithm from the Brain Connectivity Toolbox[120–122] to group time-series signals (nodes) to communities (groups of nodes) until the maximum possible score of $Q$ has been obtained. The modularity estimates for a given network is, therefore, a quantification of the extent to which the network may be subdivided into communities with stronger within-community than between-community connections.

$$Q_T = \frac{1}{\nu^+} \sum_{ij} \left( w_{ij}^+ - u_{ij}^+ \right) \delta_{M_i M_j} - \frac{1}{\nu^+ + \nu^-} \sum_{ij} \left( w_{ij}^- - u_{ij}^- \right) \delta_{M_i M_j}$$ (17)

where $\nu$ is the total weight of the network (sum of all negative and positive connections), $w_{ij}$ is the weighted and signed functional connection between signals $i$ and $j$, $u_{ij}$ is the strength of a connection divided by the total weight of the network, and $\delta_{M_i M_j}$ is set to 1 when regions are in the same community and 0 otherwise. '+' and '−' superscripts denote all positive and negative connections, respectively. Since the community detection algorithm is nondeterministic[123], 1,000 iterations of the network partitions were estimated for an intermediate value of the model ($\beta = 0.5$; $\sigma = 0.5$) while sweeping the structural resolution, $\gamma$, parameter between 0.5–2.0 ($\gamma$ tunes the strength of the null model: larger values identify smaller communities and v.v.). We tested the stability of the resultant partitions by calculating the normalised mutual information between the community assignments across iterations and found that $\gamma = 1.05$ provided the most stable community assignments. Using this setting, the Louvain algorithm was then applied to correlation matrices from across the parameter space and empirical recordings for consistency.

The participation coefficient quantifies the extent to which a node connects across all detected communities. This measure has previously been used to characterize diversely connected hub neurons within cortical brain networks, e.g., see[124]. Here, the participation coefficient, $PC$, was calculated for each node of our networks, where $\kappa_{isT}$ is the strength of the positive connections of node $i$ to node in

community $s$, and $\kappa_{iT}$ is the sum of strengths of all positive connections of node $i$. The community affiliation was determined following a consensus partition was created across the whole range using the 'consensus_und.m' script from the Brain Connectivity Toolbox for consistency. Briefly, this approach involves calculating the mutual information between community affiliation vectors (i.e., the 'agreement' matrix), and then identifying the most stable summary of these vectors before applying the Louvain algorithm to the matrix (with $\gamma = 1.05$). This resulted in a single consensus community affiliation vector, which we used for the subsequent estimate of the participation coefficient. The participation coefficient of a node is close to 1 if its connections are uniformly distributed among all the communities and 0 if all of its links are within its community:

$$PC_i = 1 - \sum_{s=1}^{n_M} \left( \frac{\kappa_{isT}}{\kappa_{iT}} \right)^2.$$ (18)

We report the summary mean participation coefficient, $\langle PC_i \rangle$ of the network in the paper averaged across 100 iterations of the Louvain algorithm (due to the algorithms stochasticity). This measure has previously been linked to both intransitive[21] and transitive[125] signatures of consciousness.

**Integrated Information.** Integrated information, $\Phi$, is defined theoretically as the amount of information a system generates as a whole, above and beyond the amount of information its parts independently generate[1]. Due to the complexity of the system and a large number of simultaneous activities analysed, the calculation of integrated information is typically considered to be computationally intractable. Thus, we utilised an approximated measure, $\Phi^*$, calculated through mismatched decoding developed from information theory, see ref. 19 for a full derivation of the method. Furthermore, we were still required to decrease the variables of the system and thus coarse-sampled the spatially dependent activity into 100 non-overlapping subsets, sampled following spatial dependence as if recorded using a $10 \times 10$ multielectrode array, whereas all referenced electrodes were utilised for the ECoG and EEG data. Briefly, $\Phi^* = I - I^*$, where $I$ is the mutual information that the current state of the whole system has about its past at a time-lag of $\tau = 15$ ms, and $I^*$, where disconnected $I$, is the mismatched information that cannot be partitioned into independent parts. For our analysis, we considered the most straightforward partition scheme, the atomic partition, in which $\Phi^*$ is calculated assuming each channel is independent. In this sense, the 'atomic partition' gives the upper-bound of $\Phi^*$, because it quantifies the amount of information loss ignoring higher-order interactions for decoding. Finally, within our model $\tau = 15$ ms was selected as it resulted in the largest $\Phi^*$, across $\tau = 1$ to 500 ms.

**Information theoretic measures.** Active Information Storage and Transfer Entropy are calculated across randomly samplings from the neuronal network (1% random sampling repeated 100 times), using the JIDT software package[64] and a discrete variable estimator with a timeseries history length, $1 \leq k \leq 10$, selected in order to maximise bias-corrected active information storage[93,94].

**Low-dimensional energy landscapes.** Following methodology from previous work[60], we formulate an energy landscape by first computing a one-dimensional measure of trajectories on our neuronal firing rate activity, namely the mean-squared-displacement, which is defined as

$$MSD_{t,\tau} = \left\langle \left| n_{t+\tau} - n_t \right|^2 \right\rangle_k$$ (19)

averaged over all $k$ neurons sampled from the network. The probability of observing a given MSD across the entire timeseries was then

 

calculated using a Gaussian kernel density estimation

$$P(MSD,t) = \frac{1}{4N} \sum_{i=1}^{n} K\left(\frac{MSD_{t,i}}{4}\right) \qquad (20)$$

where $K(u) = \frac{1}{2\sqrt{\pi}} e^{-\frac{1}{4}u^2}$. As is typical in statistical mechanics the energy of a given state, $E_\sigma$, and its probability are related by $P(\sigma) = \frac{1}{Z} e^{-\frac{E_\sigma}{T}}$ where $Z$ is the normalisation function and $T$ is the scaling factor equivalent to temperature in thermodynamics. In our analysis $\sum_\sigma P_\sigma = 1 \rightarrow Z = 1$ by construction and we can set $T = 1$ for the observed data. Thus, the energy of each MSD at a given time-lag $t$, $E$ is then equal to the natural logarithm of the inverse probability, $P(MSD,t)$ of its occurrence

$$E = \ln\left(\frac{1}{P(MSD,t)}\right). \qquad (21)$$

**Dynamic range.** To probe the information processing properties of model given stimuli, we calculated the dynamic range, $\Delta_s$, from the range of discernible responses to the range of stimulatory input. We calculated the response function, $F$, as $F = \sum_{t=1}^{T} n(t)$, that is the spiking activity generated over $T$, where $T = 10$ s (results are robust for varying $T = 100$ ms to $T = 2$ s), in response to a transient stimulus ($T/2$ s) of strength $S$, ranging from $S = 10^{-5}$ to $S = 10^{0.5}$ ms$^{-1}$, where the stimulus is modelled as afferent spikes generated as a Poisson process to each node at a stimulus rate $S$. Finally, $F$ was averaged across 20 trials for each stimulus intensity. After calculating the average elicited response, F, the dynamic range, $\Delta = 10\log_{10} \frac{S_{0.9}}{S_{0.1}}$, was calculated as the stimulus range (in dB) where variations in $S$ can be robustly coded by variations in $F$, after discarding elicited responses that are too small to distinguish from baseline, $F_0$, or network saturation, $F_{\max}$[70]. The stimulus range $[S_{0.1}, S_{0.9}]$ is calculated from the elicited response range $[F_{0.1}, F_{0.9}]$, where $F_x = F_0 + x(F_{\max} - F_0)$ which is the standard range reported[70,71,126,127].

**Particle-swarm optimisation algorithm.** The optimal fitting between the empirical and model state space ($\sigma,\beta$) complex, adaptive signatures ($KC,PC,\Phi^*$), was calculated by minimising the following function for each recording epoch

$$F(\sigma,\beta) = \sum_x \frac{X_{mdl}(\sigma,\beta) - X_{emp}}{X_{emp}}, \qquad (22)$$

where $x$ is the set of complex, adaptive signatures utilised for the fitting and the function is normalised by the empirical measure to allow comparison between measures with different ranges. All results in the manuscript use $x \in (KC,PC,\Phi^*)$ and Supplementary Figure 3 includes the power spectrum slope (estimated using a fast-Fourier transform between 2-50 Hz), and Supplementary Figure 4 looks at a subset of the signatures. The minimisation function first releases a swarm of 100 particles randomly bound within the model ($\sigma,\beta$) state-space (*particleswarm* function MATLAB) and then performs a local interior-point nonlinear convex optimisation (*fmincon* function MATLAB).

**Connected burst cascades.** To quantify coordinated spatiotemporal burst activations we detected these patterns based on their spatial and temporal contiguity, with each of them being referred to as a cascade[128]. Specifically, for successive timesteps, neuronal action potentials defined as bursts ($c = -55$, and $d = 4$) are clustered within a radius $r_S$, and a cascade is defined as a spatiotemporally contiguous set of bursts within a radius $r_T$ between successive timesteps. We present results for $r_S = 10 \cong d_E$, i.e., bursting neurons are causally excitatory connected and $r_T = 2$ ms, i.e., the bursts are binned into windows of

2 ms. The results are robust across $r_S = 5$ to 30, and $r_T = 0.5$ to 5 ms; however, larger values decrease the number of detectable patterns and smaller values result in unitary cascades. This method allows multiple simultaneous cascades to be detected. A cascade is quantified by two quantities: size, S, the number of bursts within a cascade and duration, T, the number of successive timesteps a cascade is active.

## Reporting summary
Further information on research design is available in the Nature Portfolio Reporting Summary linked to this article.

## Data availability
The data generated in this study are provided in the Source Data file. The macaque ECoG data[9] can be obtained from (www.www. neurotycho.org/sleep-task). The human EEG data[8] can be obtained following approval by the institutional review board at the University of Wisconsin-Madison. Source data are provided with this paper.

## Code availability
The model and analyses that support the findings of this study are available on GitHub (https://github.com/Bmunn/Layer5_Arousal).

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

## Acknowledgements

This project was supported by the National Health and Medical Research Council (GA294929).

## Author contributions

B.R.M.: Conceptualisation, Methodology, Investigation, Data Curation, Writing—Original Draft. E.J.M.: Writing—Review & Editing, Methodology. S.L.N.: Writing—Review & Editing, Funding. J.T.L.: Writing—Review & Editing Methodology. V.M.: Investigation, Methodology, Data Curation. R.S.: Data Curation, Writing—Review & Editing. J.M.S.: Conceptualisa- tion, Methodology, Writing—Original Draft, Supervision, Funding.

## Competing interests

The authors declare no competing interests.
