## [Peer Review File · Nature Communications]

REVIEWER COMMENTS

Reviewer #1 (Remarks to the Author):

The authors used computational models of spiking neuronal networks to explore the link between states of consciousness and network activity parameters such as coupling/burstiness and spatial coordination between neurons. While the question is interesting, the results primarily try to validate previous theory rather than experimental data and often lack sufficient detail to evaluate the soundness of the arguments. In addition, it is not clear if the results and conclusions add significantly to the understanding of the underlying mechanisms of brain states. I therefore think that the manuscript requires a significant amount of further work and is unsuitable to Nature Communications.

Major concerns:

1. The results are described mostly in a qualitative manner, lacking supporting detailed values and statistics to support the statements.
For example, the analysis of the match between experiment and model is unclear (e.g. Fig 2E, lines 251 – 261). The authors seem to be using one or more of their three complexity measures, but no values are given for experiment and model to evaluate the quality of the fit and the analysis.
2. Are common measures of EEG analysis such as the power in certain frequency bands of the EEG informative for state separation in the experiment and model? How do they compare with the measures the authors chose to use?
3. Features of sleep/awake states can overlap (as seen in the EEG/ECoG data in Fig 2d,e), yet the authors don't address this aspect. Their parameters space inverted solutions seem more separated than the actual data is.
4. Except for figure 4, results lack example raster plots of firing in the network, or simulated EEG, and thus make it hard to evaluate the model analysis and results.
5. A weakness of the study is that most of the modeling results try to validate previous theory, rather than (except for figure 2) comparing model and experimental data. Also, the mechanisms that the authors propose to explain existing theory do not seem to add much to what previous studies already indicate.
6. Results involve elaborate text that mostly belongs in methods (e.g. line 134 - 147) and introduction/discussion (e.g. line 152-155, line 212-222).

Reviewer #2 (Remarks to the Author):

Review Comments

Summary:

This work concerns the introduction of a biophysical model with the aim to investigate how three key metrics associated with consciousness manifest in states of wakefulness and low arousal (sleep, anesthesia). The authors use a spatially embedded network model of simulated neurons, each consisting of an apical and a basal dendritic compartment. They hypothesize that the activity of coordinated bursting in populations of LVPN should recreate key macroscopic signatures as those met during conscious awareness, i.e. complexity, topological integration, and integrated information.

To examine the effects of arousal on the LVPN, they altered two model parameters:

1. the amount of apical-to-basal dendritic coupling (β), which controls the apical-somatic electrotonic threshold required for the apical activity to transition the somatic compartment into burst firing (ranges $\beta = 0$, i.e. LVPN cannot burst, to coupled $\beta = 1$ where all neurons burst).
 2. the spatiotemporally connected bursting (σ), which captures the spatiotemporal coordination of bursting (ranges from spatiotemporally disconnected bursting $\sigma = 1$ to connected bursting $\sigma = N$).
- By a combination of β and σ , the authors could define different states and the nonlinear network model of LVPN could reproduce differential scales of activity – from microscale neuronal spiking to coarse-grained population activity, which they later contrasted with human and macaque empirical data across different states of arousal.

They found that in the awake state:

- there was a generalised increase in complexity proportional to the coupling between apical and somatic dendrites.
- the maximal complexity occurred with connected bursting
- an increase in topological integration
- that the mixture of both regular spikes but with spatiotemporally coordinated bursting ($\sigma \rightarrow N$) of LVPN leads to an increase in the integrated information

To investigate how these states relate to the empirically observed activity across arousal states, they further analysed ECOG recordings from 2 macaque monkeys as they transitioned from natural sleep to awake. In both monkeys, the neural dynamics increased in complexity, integration, and integrated information with arousal. Also, they analysed human EEG recordings under Dexmedetomidine anaesthesia, an agent which indirectly decreases noradrenergic levels in the thalamus and cerebral cortex, thus impairing inter-compartmental coupling in LVPN.

Then they inverted the model to identify the neuronal dynamics that differentiate unconscious, dreaming, and awake arousal states and show that the awake regime displayed rich interacting long-lasting cascades, whereas dreaming consisted of transient, localised large cascades, which facilitated optimal information processing (AIS, and TE higher in awake). Importantly, they show that the higher disconnected bursting state could be reached by targeted anatomical projections from the cholinergic nucleus basalis of Meynert. The connected bursting state could be reached by diffuse projections from the adrenergic locus coeruleus

The authors conclude that these burst cascades may reflect a fundamental unit of communication that binds information processing across widespread areas within the cortex, which is how the brain works during conscious processing. They interpret their findings as an established bridge between theoretical and empirical data as the AIM model fits their simulations

Evaluation

Overall, I find that the paper is clearly written, highly educational, exciting to read, with sound argumentation leading to the working hypothesis. The paper presents a plethora of results, but the narrative form eases the reading and understanding. Its originality lies on the study of active bursts, which is difficult to explore in vivo due to the lack of super-high resolution recordings. In a nutshell, a very accessible and important work.

Please, see below some comments that may help clarify some points in the ms

- I would suggest that notions which seems to be used interchangeably used in the ms to be differentiated. Specifically, "Conscious state", "awake state", "conscious processing", and "awareness". In my understanding, this work deals mostly with awake and non-waking states which are the pre-requisite for conscious awareness to happen, and not so much about awareness.
- How can we know that these selected signatures are (jointly) sufficient to recreate macroscopic

signatures? Why not other combination of metrics?

- Is there a control network which would not respond in these way if its parameters are tuned in the same way? Can there be such a null model?
- Since the AIM model is mentioned as a framework in the Introduction, can we claim that one of the purposes of the paper was to verify/negate this framework?
- Are the data open for review and sharing?

Minor

- Line 391: please specify that TE is a directional measure to differentiate from mutual information which is not
- Abstract: please, mention that it's a network "biophysical" model
- Introduction: I think that the AIM model is better described in Hobson, J. A., Pace-Schott, E. F., & Stickgold, R. (2000) Behavioral and Brain Sciences, 23(6), 793–842., rather than Pace-Schott, E. F. & Hobson, J. A. (2002) Nat Rev Neurosci.
- Discussion: Line 516: "In contrast to previous approaches(65)"; please shortly provide what they did and were not efficient

- Figure 1: please, provide a Note to define the acronyms mentioned in Figure panels. The green-blue colours of panel g are not clearly distinctive, another color may help with a better visualization.
- Figure 2e&f: It's recommended to use different types of teal color/shading to indicate what refers to monkey sleep and what to human anesthesia states

- Figure 5: "Conversely, relative to baseline ACH preferences ongoing dynamics (no change) making large changes unlikely (high energy).": a verb seems to be needed

Thank you for this important and fine work.
I remain at your disposal for any clarifications.

Sincere regards
Athena Demertzi,

Reviewer #3 (Remarks to the Author):

The formatting is in markdown. This helped you to visualize some of the equations.

Summary

Munn and colleagues employ a multiscale modeling approach to link single-neuron dynamics to macroscopic, coarse dynamics found in vivo via ECoG and EEG. The authors' model of Layer 5 pyramidal neurons features two compartments and an input to each neurons apical dendrite that is shared and correlated between neighbours. They formalize this using two control parameters (the strength of apex-soma coupling β , and the degree to which apical inputs are correlated σ), which they convincingly relate to neuromodulators relevant for arousal states.

By identifying those parameter combinations that best reproduce empirical ECoG and EEG observations, the authors find non-overlapping (β , σ) regions that resemble different arousal states. Using this distinction allows for a mapping between the (β , σ) space and a more abstract Attention-Input-Modulation space, which is then used to illustrate how neuromodulators would explain arousal-state changes in an analogy of state-space to a free-energy landscape.

Further, the distinction of parameter regions representing arousal states enables the authors to characterize the respective neuron dynamics. In particular, the awake state is found to comprise of causally "connected burst cascades" where subpopulations of neurons transiently enter high-firing states, which the authors show to maximize information processing, compared to the dynamics of lower-arousal states.

Overall, the paper is well written and convincing. Most modeling choices and conclusions are well motivated and connected to the literature, and the results are presented in a logical way. Figures are superb, with tremendous attention to color choices and details. Thus, the majority of our comments is of technical nature. I see a few main areas where the manuscript can improve:

- How robust are results, in particular with respect to the choice of order parameters (KC, PC, Φ)?
- Methods require some clarifications and consistency.
- Terminology could be more consistent.
- Importantly, "raw" data should be shown more extensively, to be transparent wrt to the emerging model dynamics.

Below, we provide more details.

Robustness wrt chosen order parameters, and observables

- It seems, the model inversion finds (β , σ) so that, after coarsening the dynamics, the order parameters KC, PC and Φ are matched as closely as possible between the model and EEG/ECoG recordings. Since the model inversion, and, thus, the identified regions in (β , σ)-space depend on it, how would results change when using other order parameters, or only a subset. Please comment.
- Closely related: Are these three parameters sufficient to describe all relevant dynamics? Please add an according motivation or comment to the manuscript.
- Concerning the Participation Coefficient (PC): The authors state that detected communities change with arousal state, but they form a state-independent consensus. This raises doubt about the PC measure, as the detected communities are at the core of this scalar value. It seems futile to average across dynamic states: it is not surprising that the functional connectivity changes with arousal, and that community structure is state-dependent. Averaging will undoubtedly destroy this state-dependence to some degree.
- * What happens if communities are identified per realization (or averaged per arousal state) and PC is calculated from those differing partitions? Please comment.
- KC, PC and Φ are very high-level, abstract observables, but from the current manuscript it does not become clear how the basic dynamics of the model is shaped. How do simple observables change with β and σ ? For instance, many dynamic features (bursts) crucially hinge on the firing rate. Please elaborate.
- Also, please add a panel depicting the mean firing rate, matching Fig 2 a-c, and to indicate the mean rate for the three example rasters in Fig. 1g.
- Similarly, do time-series (traces) and power spectra match between model and EEG/ECoG data, at least qualitatively?
- In Fig 2a-c, why do the combinations of β and σ lead to the shown results? From model considerations alone, some interpretation and intuition should be possible. For instance: the independence of KC from σ in Fig 2a seems trivial due to the way KC is calculated from summing over all N. However, KC should depend on the time window chosen. Please comment, and consider amending the manuscript.

Methods

Let us start with a few methodological questions:

- Concerning the coarsening: The authors tile the populations in a perfectly non-overlapping manner, but spurious correlations can crucially impact macroscale observables. I wonder how overlapping regions in space would affect the inferred observables of information processing. Please comment.
- Concerning the Participation Coefficient: From the according methods section, it is somewhat unclear how exactly the modularity definition and the identified communities (Eq. 21) are used to calculate PC (Eq. 22). In particular: are "regions" the same as "neurons", and are "modules" the same as "communities" (~l1122)? If not, what is the difference?
- Concerning the Kolmogorov complexity, l1095: What is the time-window, or the number of bins over which the sequences are counted?
- Concerning the Integrated Information, l1143: How is the time-lag of 15ms motivated?
- Concerning the simulations: The chosen time step of 0.5 ms seems to be on the larger end of common values (Neuron simulator: 0.025 ms, Brian2: 0.1 ms). Presumably, large timesteps can lead to increased synchrony, which may be relevant here for the formation of cascading bursts. Have you ruled out this kind of effect? Please comment.

Below, a few suggestions:

- The Izhikevich model (Ref 37) as denoted here is not dimensional (l864). This is easy to see by including units in Eq. (1), leading to mismatching dimensions. This comment extends through large parts of the methods (e.g. also Eq. 8); thus please check consistency of units. Admittedly, this inconsistency is also present in the original reference, but Izhikevich denoted the model to be dimensionless and only assigned units later in his manuscript. I suggest to either consider the quantities to be unitless (as is the case in the authors source code) or to use the alternative form of Izhikevich 2007 (Dynamical Systems in Neuroscience):
$$\dot{v} = k \left(v - v_{\text{r}} \right) \left(v - v_{\text{t}} \right) - u + I \quad \text{if } v \geq v_{\text{peak}}$$
$$\dot{u} = a \left(b \left(v - v_{\text{r}} \right) - u \right) \quad \text{if } v < v_{\text{peak}}$$
$$c \rightarrow u, \quad u \rightarrow d$$
- In l883ff, what is the neuron density in the model? Without this information, the specified $200 \mu\text{m}$ connection radius are arbitrary. In more practical terms, what is the measured / expected degree? I suggest to adapt a consistent spatial notation, matching how the Mexican-hat and Gaussian smoothing kernel are expressed (in units of neurons l 907, also l 926). Note also a typo in the inline math "N(6)".
- In the results section, l 323, we were unable to find the referenced methods section on how the cluster identification is performed. Please point out the relevant section and make the link clearer.
- From Eq. 7 and surrounding text, it is unclear to me whether $I_{\text{ap}}(t)$ is a global variable, independent of neuron id (as of the equation) or if it is missing an index (j) to be neuron-specific (as of the preceding text).
- The description of model coarsening is inconsistent between l172, l1038ff, l1082. Were there 100 non-overlapping groups, or 100 neurons per group?
- In Figure 4 a-c, how were the time-series binarized to determine which voxel was bursting, regular-spiking or silent? I could not find a reference to the respective methods section.
- Concerning the source code: We greatly appreciate that the authors share their code and we hope they will make it accessible after publication. However, we think that a round of cleaning and documentation is needed in order to make it really useful — and to enable a reader to directly connect methods and their implementation.

Terminology

- In some places of the manuscript, it is hard to connect the word-level description to the corresponding variable (β , σ , KC, PC, Φ), because the descriptions change and

overlap. For instance, β is referred to as "dendrite coupling" l150 and "low/high bursting" fig 1g, l254, l259, whereas σ is referred to as "connected bursting" fig 1g, l161 or "coordinated bursting" l208. Please pick one particular wording and employ it throughout, while only using "bursting" as a description of the dynamics, not for β or σ .

- Throughout the manuscript, "coarse graining" is used. Although it is clear what the authors mean by it, the term is taken out of context: in theoretical physics, it refers to a specific technique where spins are combined iteratively, and neither overlap nor gaps exist between the sampled tiles — which is not the case for the neuroscience recordings. Please consider choosing a different term, e.g. coarse sampling.

- Potentially, LIV and LVI are not a good abbreviations (Fig 1, ff); I suggest to use "L4, L6" instead, because LIV = 54, LVI = 56, And "LI-LII/III" = 34 + 1/3 ;)

Other

- We were misled by the section on burst-cascades in Fig 3 (between Fig 2 and 4), in the sense that the main objective of the manuscript seemed to be fulfilled when the connection between arousal states and neuron dynamics was made. This made it hard to appreciate the explanation provided by the analysis of free-energy landscapes that followed. As the connected-bursting seems somewhat more independent from the transitions between states, I suggest to keep Figs 2 and 4 together by moving Fig 3 and according text.

- In Fig 3 c, some colors are swapped: Green used to be unconscious, and blue used to be dreaming.

- In ~ ll 293ff there seems to be a slight mixup of color referencing.

- I would like to point out some additional references that might prove useful:

- Izhikevich 2007, Dynamical Systems in Neuroscience, 978-0-262-09043-8

- Concerning coarsening: Neto et al. 2022, doi 10.1371/journal.pcbi.1010678

- Concerning information flow and the breakdown thereof under anaesthesia: Breakdown of local information processing may underlie isoflurane anesthesia effects: Wollstadt et al. 2017, doi 10.1371/journal.pcbi.1005511

- Regarding mechanisms of information routing: Flexible information routing by transient synchrony: Palmigiano et al. 2017, doi 10.1038/nn.4569

Although the list of points may seem long, we hope they do help to clarify this interesting manuscript further!

We thank the three reviewers for their constructive comments, which have led to a number of changes in the manuscript that we have highlighted in blue. Below, we detail our response to each of the reviewers concerns.

Reviewer #1 (Remarks to the Author):

Major concerns:

1. The results are described mostly in a qualitative manner, lacking supporting detailed values and statistics to support the statements. For example, the analysis of the match between experiment and model is unclear (e.g. Fig 2E, lines 251 – 261). The authors seem to be using one or more of their three complexity measures, but no values are given for experiment and model to evaluate the quality of the fit and the analysis.

We appreciate your feedback. In the updated manuscript, we have included the detailed fitting metrics and statistics necessary to support our statements. To address this, we have expanded Figure 2 to display neuronal properties across the model parameter space, giving a clearer picture of how the model performs under different parameters, as well as the coarse-sampled dynamics.

The following text was added to p7: *“We explored how the multiscale neural dynamics differ from neuron to coarse-sampled across the model state space. At the neuronal scale, the model state-space displays a smooth increase in mean firing rate, with apical-basal compartment coupling (β) consistent with the increase in bursting (Fig. 2a). To further quantify this dynamical heterogeneity in spiking variability dependent upon the state parameters we calculated two standard empirical neuronal measurements. The spike-count Fano-factor (FF; variance/mean), which quantifies the mean-normalised firing rate variability, increased with β peaking at maximal apical input correlation (σ) and intermediate-high values of β but decreases again as β trended towards unity, suggesting that the network became saturated once bursting was too prevalent (Fig. 2b). Furthermore, we explored how apically-*

driven input may alter emergent correlations between neuronal spiking activity. We found that intermediate β was associated with an elevated, albeit low mean pairwise spike-count correlation ($r_{SC} : \langle r_{SC} \rangle < 0.07$; Fig. 2c), consistent with experimental predictions⁴¹. These results demonstrate that a simple dual-compartment model reciprocally connecting nonlinear spiking neurons is capable of supporting substantial heterogeneous spiking dynamics.”

Furthermore, to demonstrate the accuracy of the fitting process, we have included the quality of fits for the three parameters in all recordings in the Supplementary figures, allowing readers to evaluate the quality of the fit between our model and experimental data.

For Macaque ECoG:

For Human dexmedetomidine:

We have also included how the fits change when using only two of the three parameters, providing additional insights into the relationship between the model and the data. We believe that these additions provide the necessary details and statistics to support our statements and improve the overall clarity of the manuscript.

“**Fig. S4:** Comparison of model fits on a macaque recording (Fig. 3a) only using two complex, adaptive signatures. **(a)** Model state-space (left) and match with empirical measurements (right) after removing KC. **(b)** Model state-space (left) and match with empirical measurements (right) after removing PC. **(c)** Model state-space (left) and match with empirical measurements (right) after removing Φ^* .”

The following text was added to p11: “We found that the model-inverted sleep, waking, and awake parameters are localised and statistically distinct ($p < 0.01$; Fig. 3c). The teal regions in Figure 3c demonstrate that the empirical sleep dynamics are optimally-matched with simulated coarse-sampled dynamics of $L5_{PN}$ with low coupling ($\beta < 0.2$). Waking moved the matched dynamics through a period of increased coupling (Fig. 3c yellow), and the awake regime was dominated by transient spatially correlated bursting (Fig. 3c red). Crucially, sleep and awake are significantly different regions of the model parameter space ($p < 0.001$). The optimisation swarm-fitting results in a tight match between model and empirical complex, adaptive signatures model-matched measures are within 10% of empirical values (Fig. S1).”

2. Are common measures of EEG analysis such as the power in certain frequency bands of the EEG informative for state separation in the experiment and model? How do they compare with the measures the authors chose to use?

In the original manuscript, we chose not to use traditional frequency band measures in our analysis due to the fact that changes in spectral power and mean-field theories have been extensively studied (and even used for biological mapping between data and theory – Jirsa and Haken, Field theory of electromagnetic brain activity, PRL, 1996; Wilson and Cowan, A mathematical theory of the functional dynamics of cortical and thalamic nervous tissue, Kybernetik, 1973; Robinson et al., Steady states and global dynamics of electrical activity in the cerebral cortex, PRE, 1998; Abey Suriya and Robinson, Real-time automated EEG tracking of brain states using neural field theory, J Neurosci. Methods, 2016).

Nevertheless, to ensure our method reproduces broadscale spectral properties, we have additionally explored spectral slope as a parameter in the model and found it to be strongly correlated with complexity.

We then tested the model's predictions against empirical recordings from macaque monkeys and found that the model matches the empirical slope, indicating that it captures an important feature of brain activity. However, we opted to not include spectral slope as a fitting parameter in our model fitting approach because it did not significantly change the quality of the fits.

Comparison of model fits – left, without spectra as a parameter; right, with spectra as a parameter:

Comparison of x/y fits between the two:

Overall, the measures used in our study were specifically chosen *a priori* to capture the complex and dynamic nature of brain activity that we aimed to investigate in this study. While frequency band measures can provide important insights into brain activity, we

believe that our chosen measures better align with our research questions, desire to explore novel macroscale measures, and the nature of our model.

The following text was added to p9: *“Importantly, these three complex, adaptive signatures were selected due to their empirical use and to ensure a diverse set of analytical approaches: for example, KC and Φ^* reflect univariate and multivariate information theoretic measures, while PC is a topological measure. Their differentiation across the model parameter space emphasises their utility to distinguish differential coarse-sampled dynamics. Another typically utilised signature of macroscale arousal is spectral band-limited power, however, this was not a focus of the study, due to the extensive existing mean-field theoretical studies and empirical fitting linking empirical spectra to thalamocortical resonances (see⁵⁵⁻⁵⁷). For completeness, we calculated the spectral slope (Fig. S2a) – an indicator of arousal – across the model state-space and found a flattening of the spectral slope with increasing β coupling between apical and basal dendritic compartments. That is to say: coupling mediated bursting increases high-power and flattens the spectral slope in a way that was tightly correlated with signal complexity (KC).”*

The following text was added to p11: *“As the swarm-fitting algorithm is limited in its capacity to discriminate regions by the properties of the complex, adaptive signatures we explored including the spectral slope as a further fitting parameter. Consistent with the other measures the spectral slope varied across all three arousal states in the macaque recording (Fig. S1; $p < 0.05$ KW) and the updated fitting led to a subtle change in fitting values. Furthermore, removing parameters and repeating the fitting procedure led to differential locations, in particular Φ^* , suggesting three metrics offer a unique discrimination.”*

3. Features of sleep/awake states can overlap (as seen in the EEG/ECOG data in Fig 2d,e), yet the authors don't address this aspect. Their parameters space inverted solutions seem more separated than the actual data is.

We apologize for any confusion caused by the perceived separation of sleep/awake states in our parameter space inverted solutions. To address this, we have included an additional figure in the manuscript that demonstrates the spread of arousal across the three parameters. This figure shows that there is already a large spread of arousal across the parameter space, and we believe that this helps to capture the variability and overlap of sleep/awake states.

We have also added further text to the manuscript on p10: “*We next asked whether the different arousal dynamics (3-dimensional signatures) would coincide with our model’s predicted activity. To test this hypothesis, we utilised a hybrid optimisation approach to minimise the summation of the absolute relative difference between empirical and simulated complex, adaptive dynamics across each epoch (see Methods). That is to say, we wanted to find the mapping from the 3-dimensional signatures (Fig. 3b) to the 2-dimensional model state space (Fig. 3c). To achieve this we utilised a population-based stochastic particle swarm⁴⁸ to search the model parameter space before a fine-scale, interior-point nonlinear convex optimization was deployed to ensure the discovery of a precise global minimum (see Methods).*”

The following text was added to p11: “*Figure 3c demonstrates that the awake state complex, adaptive dynamics in non-human primate is unlikely to be explained by dynamics: without bursting ($\beta < 0.2$; $p < 0.001$); with only bursting ($\beta > 0.8$; $p < 0.001$); or with spatially uncorrelated bursting ($\sigma < N/4$; $p < 0.001$). Results are consistent when calculated on a separate monkey with sleep clearly distinguishable from waking and awake states. However, the waking and awake states overlap at the later stages suggesting the animal awakens faster in this recording (Fig. S5).*”

4. Except for figure 4, results lack example raster plots of firing in the network, or simulated EEG, and thus make it hard to evaluate the model analysis and results.

Thank you for your feedback. We agree that it is important to provide examples of the simulated neural activity and EEG data to support the results and analysis presented in the manuscript. To address this, we have included additional figures that provide a more detailed visualization of the neural activity and EEG signals generated by the model.

The updated version of Figure 1, complete with simulated coarse-sampled signals and raster plots.

We have also included a summary plot in Figure 2 describing standard univariate/multivariate measures of the models' simulated neuronal activity underlying the coarse-sampled activity:

“Figure 2. Multiscale modelling allows the comparison between neuronal activity (top) and paradigmatic macroscale signatures of complex, adaptive dynamics (bottom) in the coarse-sampled activity across the model state space. (a) mean neuronal firing rate; (b) Mean spike-count Fano-factor (FF; variance/ mean); (c) mean Pearson spike-count pairwise correlation $\langle r_{SC} \rangle$; (d) mean Kolmogorov complexity (KC); (e) mean participation coefficient (PC); and (f) integrated information Φ^ .”*

5. A weakness of the study is that most of the modelling results try to validate previous theory, rather than (except for figure 2) comparing model and experimental data. Also, the mechanisms that the authors propose to explain existing theory do not seem to add much to what previous studies already indicate.

We apologize for the confusion regarding our modeling results. Our intention was not to validate the AIM model or any previous theory, but rather to explore the link between neuronal dynamics and mesoscale activity in a data-driven approach. We have revised the manuscript to make this clearer and have moved the qualitative comparison with the AIM model to the end of the paper.

Regarding the lack of comparison between model and experimental data; in the revised manuscript, we have expanded Figure 1 to include spiking rasters and coarse-sampled activity, and Figure 2 now displays neuronal properties across the model parameter space. Furthermore, we have included quality-of-fit measures for the three parameters in all measurements in the supplementary figures to demonstrate the fitting process.

Finally, we acknowledge that previous studies have already indicated aspects of these mechanisms. However, our study adds value by linking neuronal dynamics to mesoscale activity that has been missing from previous work. We hope this addresses your concerns and thank you again for your feedback.

6. Results involve elaborate text that mostly belongs in methods (e.g. line 134 - 147) and introduction/discussion (e.g. line 152-155, line 212-222).

We appreciate the feedback and agree that some of the text could be better organized. Our intention was to provide the reader with sufficient detail to appreciate the complexity of our model and understand the main findings neuronal dynamics – mesoscale complex adaptive signatures. In addition, we note that this issue refers to a stylistic choice associated with the journal in question (methods after results), rather than with our explicit choice to include information this way. As such, although we have aimed to streamline the results section and moved some of the more detailed explanations to the results, we have retained the basic overall structure of the manuscript.

Reviewer #2 (Remarks to the Author):

Evaluation

Overall, I find that the paper is clearly written, highly educational, exciting to read, with sound argumentation leading to the working hypothesis. The paper presents a plethora of results, but the narrative form eases the reading and understanding. Its originality lies on the study of active bursts, which is difficult to explore in vivo due to the lack of super-high resolution recordings. In a nutshell, a very accessible and important work.

We thank the reviewer for their time and effort reading the manuscript, as well as their kind assessment of the work.

Please, see below some comments that may help clarify some points in the ms.

- I would suggest that notions which seems to be used interchangeably used in the ms to be differentiated. Specifically, “Conscious state”, “awake state”, “conscious processing”, and “awareness”. In my understanding, this work deals mostly with awake and non-waking states which are the pre-requisite for conscious awareness to happen, and not so much about awareness.

Thank you for your suggestion to differentiate between the terms "Conscious state", "awake state", "conscious processing", and "awareness". We have revised the manuscript by removing the phrases "conscious processing" and "awareness", and utilized the "awake state" and "conscious state" terminology to better reflect the brain states discussed in the manuscript. We believe these revisions have improved the clarity of the manuscript.

- How can we know that these selected signatures are (jointly) sufficient to recreate macroscopic signatures? Why not other combination of metrics?

Thank you for your thoughtful comment on our manuscript. We appreciate your question regarding the selection of parameters and their sufficiency to recreate macroscopic signatures. We chose the three selected parameters as they each represent a typical signature of arousal states.

The following text was added to p9: *“Importantly, these three complex, adaptive signatures were selected due to their empirical use and to ensure a diverse set of analytical approaches: for example, KC and Φ^* reflect univariate and multivariate information theoretic measures, while PC is a topological measure. Their differentiation across the model parameter space emphasises their utility to distinguish differential coarse-sampled dynamics. Another typically utilised signature of macroscale arousal is spectral band-limited power, however, this was not a focus of the study, due to the extensive existing mean-field theoretical studies and empirical fitting linking empirical spectra to thalamocortical resonances (see ^{55–57}). For completeness, we calculated the spectral slope (Fig. S2a) – an indicator of*

arousal— across the model state-space and found a flattening of the spectral slope with increasing β coupling between apical and basal dendritic compartments. That is to say: coupling mediated bursting increases high-power and flattens the spectral slope in a way that was tightly correlated with signal complexity (KC).”

To address your concern, we have also conducted additional analyses with less parameters, which resulted in slightly different findings, and more parameters, which altered the model-matched locations, yet preserves the global separation (contrast Fig. 3a with Fig. S3). We believe that including more varied signatures would likely improve the fitting capacity of the model.

“Fig. S4: Comparison of model fits on a macaque recording (Fig. 3a) only using two complex, adaptive signatures. (a) Model state-space (left) and match with empirical measurements (right) after removing KC. (b) Model state-space (left) and match with empirical measurements (right) after removing PC. (c) Model state-space (left) and match with empirical measurements (right) after removing Φ^* .”

The following text was added to p11: “Consistent with the other measures the spectral slope varied across all three arousal states in the macaque recording (Fig. S1; $p < 0.05$ KW) and the updated fitting led to a subtle change in fitting values (Fig. S3). Furthermore, removing parameters and repeating the fitting procedure led to differential locations, in particular Φ^* , suggesting three metrics offer a unique discrimination (Fig. S4).”

The following text was added to p20: “Nevertheless, we expect further dynamical signatures would further carve out the model’s parameter space and improve the fitting procedure for empirical data.”

Is there a control network which would not respond in these way if its parameters are tuned in the same way? Can there by such a null model?

We agree that any network controlled through gain augmentation will likely show similar signatures (e.g., kuramoto model), but the crucial question for us is whether the non-linear effects of $L5_{PT}$ relate to the signatures we observe across arousal states. In this case, it is an intractable problem to rule out all potential alternative generative null models. Instead, the goal of our project was to prove that this model was sufficient to reproduce the dynamics observed across arousal.

Since the AIM model is mentioned as a framework in the Introduction, can we claim that one of the purposes of the paper was to verify/negate this framework?

Thank you for your comment regarding the AIM model mentioned in our manuscript. We want to clarify that the purpose of our study was not to verify or negate the AIM model. The AIM model was discovered after our initial analysis, and we only referenced it as a framework in the introduction. To avoid any further confusion, we have moved the discussion of the AIM model to the end of the manuscript (new Fig. 6). We hope this clarification addresses your concern.

Are the data open for review and sharing?

The macaque data is open and the model code is available on GitHub (https://github.com/Bmunn/Layer5_Arousal). Unfortunately, we are unable to make the human EEG data open source.

Line 391: please specify that TE is a directional measure to differentiate from mutual information which is not

We have updated the manuscript to include this information.

Abstract: please, mention that it's a network "biophysical" model

We have updated the manuscript to include this information.

Introduction: I think that the AIM model is better described in Hobson, J. A., Pace-Schott, E. F., & Stickgold, R. (2000) Behavioral and Brain Sciences, 23(6), 793–842., rather than Pace-Schott, E. F. & Hobson, J. A. (2002) Nat Rev Neurosci.

Thank you for pointing this out. This information has been updated.

- Discussion: Line 516: “In contrast to previous approaches(65)” ; please shortly provide what they did and were not efficient

We have updated the manuscript to include this information.

The following text was added to p20: “In contrast to previous approaches ⁶⁶, we selectively analysed bursting neurons and defined a burst cascade as a sequence of burst-induced burst firing in a set of connected neurons, as it is empirically difficult to discriminate between regular spikes-bursts using calcium imaging and to know causal anatomical connections. ”

- Figure 1: please, provide a Note to define the acronyms mentioned in Figure panels. The green-blue colours of panel g are not clearly distinctive, another color may help with a better visualization.

We have updated the manuscript to include this information.

“Figure 1. Nonlinear layer 5 pyramidal neurons link across scales and network model architecture. (a) The ascending arousal system (AAS) and thalamus are pivotal for controlling arousal state and project to the cortex through specific (targeted) and nonspecific (diffuse) projections, such as the nucleus basalis of Meynert (nbM) and Locus coeruleus (LC). **(b)** Layer 5 pyramidal neurons dominate macroscale electrophysiology due to their large and parallel dipole moments. Thick-tufted layer 5 pyramidal neuron consists of basal (L4-L6) and apical (L1-L2/3) dendrites that are physically and electrotonically separated by the apical trunk. The two dendritic regions receive differential input across specific/ nonspecific projecting thalamic and ascending arousal fibres. **(c)** Constant current driven into basal dendrites generates regular action potentials (‘spikes’; green), whereas simultaneous activation of apical and basal dendrites can transition the cell into high-frequency burst spiking (yellow). **(d)** We simulate activity in a network of biophysical dual-compartment pyramidal neurons. **(e)** We explore two parameters the apical-basal compartment coupling (β) – controlling the electrotonic threshold required for the apical activity to transition the basal compartment into burst firing – and the spatially correlated apical compartment input (σ) – whether bursts can occur between adjacent and reciprocally connected neurons, which are modified by subcortical structures. **(f)** Sub-panels denote different regions of the state space (model parameters) leading to different combinations of (top) neuronal spiking (green) and

bursting (yellow), and (bottom) coarse-sampled population activity for identical system input – i) low β and intermediate σ ; ii) intermediate β and low σ ; iii) intermediate β and high σ ; and iv) high β and intermediate σ .”

- Figure 2e&f: It’s recommended to use different types of teal color/shading to indicate what refers to monkey sleep and what to human anesthesia states

We have updated the new Figure 6 in the manuscript where the measures overlap for differentiation, and we have left the colours in Figure 2 as they are differentiated by the row.

Figure 6. Multiscale modelling and matching across scales confirm theoretical predictions of the Activation/Information/Modulation (AIM) model. (a) Theoretical state space of arousal proposed by the AIM model of sleep (reproduced from ¹) where increasing activation moves from a diminished consciousness state to either an awake or dreaming state separated by ratios of cholinergic or monoaminergic neuromodulation. The theoretical state space is closely recapitulated in our empirical coarse-sampled matched model state space, moving from states of unconsciousness (sleep-macaques, dark teal; anaesthesia-humans, light teal), through intermediate consciousness (waking-yellow), to conscious states separated by disconnected (dreaming-blue) and connected bursting (awake-macaques, dark red; humans, light red). **(b)** (top-left) The model’s complex, adaptive dynamics (Fig. 2d-f) are optimally clustered into 5 clusters as assessed by the Davies-Bouldin criterion (DBC) on 500 stochastic k-means clustering. (right) Four of the five regions overlap with empirically observed states, and a fifth region coinciding with high bursting.

- Figure 5: “Conversely, relative to baseline ACH preferences ongoing dynamics (no change) making large changes unlikely (high energy).“: a verb seems to be needed ach supports sustained activity quenching variability ..

We have updated the manuscript.

The text page 15 now reads: “(c) Relative to baseline, NAd decreases the energy of all brain-state transitions (i.e., makes all state changes more likely). Conversely, ACh increases the energy of large MSD and decreases the energy of small MSD (i.e., quenches variability supporting ongoing dynamics).”

Reviewer #3 (Remarks to the Author):

Robustness wrt chosen order parameters, and observables

- It seems, the model inversion finds (β, σ) so that, after coarsening the dynamics, the order parameters KC, PC and Φ are matched as closely as possible between the model and EEG/ECOG recordings. Since the model inversion, and, thus, the identified regions in (β, σ) -space depend on it, how would results change when using other order parameters, or only a subset. Please comment.
- Closely related: Are these three parameters sufficient to describe *all relevant* dynamics? Please add an according motivation or comment to the manuscript.

Thank you for your insightful comment regarding the selection of the three parameters in our model inversion. We would like to clarify that the three parameters were not chosen to be exhaustive, but rather reflect three highly cited signatures for quantifying brain dynamics across arousal states.

We explored removing individual parameters and found that Φ and participation are important metrics for fitting along the x-axis, yet the global separation is preserved (contrast Fig. 3a with Fig. S3). We would like to note that the swarm is a stochastic fitting algorithm, and we expected subtle changes.

“Fig. S4: Comparison of model fits on a macaque recording (Fig. 3a) only using two complex, adaptive signatures. (a) Model state-space (left) and match with empirical measurements (right) after removing KC. (b) Model state-space (left) and match with empirical measurements (right) after removing PC. (c) Model state-space (left) and match with empirical measurements (right) after removing Φ^* .”

Overall, our chosen measures were specifically chosen *a priori* to capture the complex and dynamic nature of brain activity that we aimed to investigate in this study.

The following text was added to p9: *“Importantly, these three complex, adaptive signatures were selected due to their empirical use and paradigmatical reflection of diverse analytical techniques. For example, KC and Φ^* reflect univariate and multivariate information theoretic measures, while PC is a topological measure. Interestingly, their differentiation across the model parameter space emphasises their utility to distinguish differential coarse-sampled dynamics.*

The following text was added to p11: *“Furthermore, removing parameters and repeating the fitting procedure led to differential locations, in particular Φ^* , suggesting three metrics offer a unique discrimination.”*

- Concerning the Participation Coefficient (PC): The authors state that detected communities change with arousal state, but they form a state-independent consensus. This raises doubt about the PC measure, as the detected communities are at the core of this scalar value. It seems futile to average across dynamic states: it is not surprising that the `_functional_` connectivity changes with arousal, and that community structure is state-dependent. Averaging will undoubtedly destroy this state-dependence to some degree. What happens if communities are identified per realization (or averaged per arousal state) and PC is calculated from those differing partitions? Please comment.

Thank you for your feedback and suggestion regarding the Participation Coefficient (PC) measure. We agree that the community structure can vary across arousal states and that averaging may destroy some degree of state-dependence – indeed, we have published on this topic in the past (e.g., see Shine et al., 2016 *Neuron*; Shine et al., 2018 *eLife*). In addition, the Louvain algorithm is stochastic, which can lead to variation in the solutions provided (although in practice, they are typically quite stable for neuroimaging data). To account for this, we used the Louvain algorithm to detect communities, and the mean participation was calculated on clusters detected by repeating the algorithm one hundred times. However, if we do not create a consensus partition, then we are faced with a separate problem – namely, that changes in the clustering solution could render the participation coefficient scores incomparable across different model parameters. While we agree that holding the community assignment static across model iterations might potentially diminish some of the effects that we might otherwise attribute to the participation coefficient, to our mind this was a less costly choice than allowing the Louvain algorithm to vary across parameters. Note that we have adopted a similar approach before in modelling work (Müller et al., 2020 *Nature Communications*), and found it to be the most parsimonious means for characterising the topology of the functional communities that arise from our approach.

- KC, PC and Φ are very high-level, abstract observables, but from the current manuscript it does not become clear how the basic dynamics of the model is shaped. How do simple observables change with β and σ ? For instance, many dynamic features (bursts) crucially hinge on the firing rate. Please elaborate.

- Also, please add a panel depicting the mean firing rate, matching Fig 2 a-c, and to indicate the mean rate for the three example rasters in Fig. 1g.

Thank you for your comment. We have revised the manuscript to address this point by including additional information about the basic dynamics of the model, specifically how simple observables change with β & σ . In Figure 1, we have modified the plots to show spiking rasters (bursts/spikes differentiated) and the coarse-sampled activity.

In Figure 2, we have included a three-parameter summary of typical neuronal measures, including the mean network firing rate (which increases with bursting), the neuronal firing rate Fano-Factor (a measure of spiking variability – variance/mean) with a peak aligning with maximal integration), and the mean neuronal pairwise correlation aligning with Φ^* . We hope that these additions provide readers with a clearer insight into the neuronal dynamics across the model parameter space.

“Figure 2. Multiscale modelling allows the comparison between neuronal activity (top) and paradigmatic macroscale signatures of complex, adaptive dynamics (bottom) in the coarse-sampled activity across the model state space. (a) mean neuronal firing rate; (b) Mean spike-count Fano-factor (FF; variance/mean); (c) mean Pearson spike-count pairwise correlation ($\langle r_{SC} \rangle$); (d) mean Kolmogorov complexity (KC); (e) mean participation coefficient (PC); and (f) integrated information (Φ^).”*

- Similarly, do time-series (traces) and power spectra match between model and EEG/ECoG data, at least qualitatively?

To address your concern, we have included the spectral slope as a further common parameter and explored how it changes the fitting in a macaque. We then tested the model's predictions against empirical recordings from macaque monkeys and found that the model matches the empirical slope, indicating that it captures an important feature of brain activity.

“Fig. S2: (a) Spectral slope across the model state space. (b) spectral slope calculated across a macaque recording (Fig. 3a).”

Comparison of model fits – left, without spectra as a parameter; right, with spectra as a parameter:

Comparison of x/y fits between the two:

As we found a close correlation between the spectral slope and the complexity signature, little change was observed in the fits, so we chose to omit it from the analysis. While frequency band measures can provide important insights into brain activity, we believe that our chosen measures better align with our research questions and the nature of our model. However, as mentioned in the manuscript (P1), the aim of the study is to investigate the relationship between macroscopic signatures of brain states and the underlying neural dynamics.

The following text was added to p9: “Importantly, these three complex, adaptive signatures were selected due to their empirical use and to ensure a diverse set of analytical approaches: for example, KC and Φ^* reflect

univariate and multivariate information theoretic measures, while PC is a topological measure. Their differentiation across the model parameter space emphasises their utility to distinguish differential coarse-sampled dynamics. Another typically utilised signature of macroscale arousal is spectral band-limited power, however, this was not a focus of the study, due to the extensive existing mean-field theoretical studies and empirical fitting linking empirical spectra to thalamocortical resonances (see ⁵⁵⁻⁵⁷). For completeness, we calculated the spectral slope (Fig. S2a) – an indicator of arousal – across the model state-space and found a flattening of the spectral slope with increasing β coupling between apical and basal dendritic compartments. That is to say: coupling mediated bursting increases high-power and flattens the spectral slope in a way that was tightly correlated with signal complexity (KC).

”

The following text was added to p11: “*As the swarm-fitting algorithm is limited in its capacity to discriminate regions by the properties of the complex, adaptive signatures we explored including the spectral slope as a further fitting parameter. Consistent with the other measures the spectral slope varied across all three arousal states in the macaque recording (Fig. S2b; $p < 0.05$ KW) and the updated fitting led to a subtle change in fitting values. Furthermore, removing parameters and repeating the fitting procedure led to differential locations, in particular Φ^* , suggesting three metrics offer a unique discrimination.*”

- In Fig 2a-c, why do the combinations of β and σ lead to the shown results? From model considerations alone, some interpretation and intuition should be possible. For instance: the independence of KC from σ in Fig 2a seems trivial due to the way KC is calculated from summing over all N. However, KC should depend on the time window chosen. Please comment, and consider amending the manuscript.

We thank the reviewer for their comment. Indeed, interpretation and intuition regarding the results in Fig 2a-c is important for a better understanding of the study. Regarding Fig 2a (now Fig.2d), we agree that the independence of KC from σ might seem trivial, but it is an important observation since KC is not independent of other parameters, as shown in Fig 3a. As for the time window chosen for calculating KC, we have calculated KC over a 1ms time-window. We have added a sentence to the manuscript to clarify this point.

The following text was added to p32: “*The population signal is calculated at the resolution of the relevant signal (i.e., $dt = 1$ ms model, monkey ECoG, and human EEG).*”

Regarding Fig 2b and c (now Fig. 2e/f), the results are more complex and we have added further explanation in the manuscript. For example, the decrease in PC with increasing β and decreasing σ (Fig 2b) is related to an increase in bursting activity, which leads to more desynchronized firing (ex. Fig. 1fii) and a reduction in the number of functional communities and ν for the increase in synchrony (matched in the peaked FF Fig. 2b). Similarly, the increase in Φ with increasing β and increasing σ (Fig 2c) is related to an increase in FF variability and neuronal mean pairwise correlation.

Overall, we agree that further interpretation and intuition could be added to the manuscript and we have amended it accordingly.

The following text was added to p8: *“The three measures varied substantially and distinctly across the model state space. We observed a generalised increase in informational capacity (complexity) proportional to the coupling between apical and somatic dendrites. This increase in information content is consistent with the increased population firing rate⁴⁴. However, we found the maximal complexity occurred with intermediate-high apical-basal dendritic coupling and spatiotemporally correlated apical input (high σ high β ; Fig 2d). Increasing apical-basal coupling increased network integration and correlated apical input (high σ) L5_{PN} led to a noticeable increase in topological integration that was most pronounced for intermediate β , where at the two extremes increasing apical-basal coupling led to an increase in network integration (Fig. 2e). The maximal integration aligns with the peaks in FF and $\langle r_{SC} \rangle$ (Fig. 2b/c).*

We next calculated integrated information (Φ^ ; Fig. 2f)¹¹. Φ^* was estimated using mismatched decoding between the coarse-sampled signals and their past at a time-lag of 15ms chosen as it led to the maximal Φ^* , consistent with empirical findings^{8,45}. Φ^* increased generally with apical-basal coupling, consistent with information complexity (KC) and integration (PC). However, we found that this measure peaked with an admixture of regular spiking and bursting aligning with the peak in neuronal pairwise correlations (Fig. 2c).”*

Importantly, these three complex, adaptive signatures were selected due to their empirical use and paradigmatical reflection of diverse analytical techniques. For example, KC and Φ^* reflect univariate and multivariate information theoretic measures, while PC is a topological measure. Interestingly, their differentiation across the model parameter space emphasises their utility to distinguish differential coarse-sampled dynamics.

Methods

- Concerning the coarsening: The authors tile the populations in a perfectly non-overlapping manner, but spurious correlations can crucially impact macroscale observables. I wonder how overlapping regions in space would affect the inferred observables of information processing. Please comment.

Thank you for your question. We appreciate the reviewers concern about spurious correlations potentially impacting macroscale observables. As suggested, we have explored the impact of overlapping regions in space on our inferred observables of information processing. To do so, we recalculated the coarse-sampling using a Gaussian smoothed sampling that simulates electrodes atop that detect overlapping signals. While we found that this does not significantly change the complexity measures (all $r > 0.9$ when contrasted with perfect nonoverlapping subsampling), it does slightly alter the functional

connectivity (likely due to a reduction in separate communities from the increased correlations) and slightly alters the Φ measure. We have included these findings in the revised manuscript supplementary figures to provide readers with a more nuanced understanding of the potential impact of spurious correlations.

We have added the following supplementary figure S1 and amended the methods.

“Fig. S1: Comparison of the three complex, adaptive signatures calculated using a spatially overlapping Gaussian smoothed coarse-sampling.”

The following text was added to p27: “Another alternative approach, aligning with empirical conditions would be to sample spiking activity with overlap across neighbouring regions¹. To do this, we subsampled spiking activity spatially pooled by convolution with a 2d gaussian with $\sigma = 3$ neurons (i.e., $40\mu m$) as if recorded using a 10×10 multielectrode array. Nevertheless, despite this significantly overlapping coarse-sampling (i.e., significant spurious correlations) the multivariate complex, adaptive signatures are only slightly affected (PC & Φ^* ; $r > 0.9$ perfectly sampled/overlapping sampling) and the univariate (KC) is unaffected (Fig. S1).”

Concerning the Participation Coefficient: From the according methods section, it is somewhat unclear how exactly the modularity definition and the identified communities (Eq. 21) are used to calculate PC (Eq. 22). In particular: are "regions" the same as "neurons", and are "modules" the same as "communities" (~11122)? If not, what is the difference?

We apologize for any confusion caused by the unclear description in the methods section regarding the use of modularity definition and identified communities for calculating the Participation Coefficient and we would like to clarify that in this context, "regions" refer to the coarse-sampled signals, and "modules" refer to the communities of these signals. We understand that these terms may have caused confusion, and we have taken steps to amend the manuscript accordingly. Thank you for bringing this to our attention.

The following text was added to p32: “The time series of activity (be that coarse-sampled population spiking activity, bipolar re-referenced ECoG, or surface Laplacian EEG), \mathbf{n} , were used to create a weighted, signed, and un-thresholded functional connectivity matrix (using the region-to-region Pearson’s correlation as a measure of functional connection strength), which we then examined for a modular network¹¹³. The algorithm optimizes a multilayer modularity quality function, Q , using a weighted- and signed- version of the Louvain modularity algorithm from the Brain Connectivity Toolbox^{114–116} to group time-series signals (nodes) to communities (groups of nodes)

until the maximum possible score of Q has been obtained. The modularity estimates for a given network is, therefore, a quantification of the extent to which the network may be subdivided into communities with stronger within-community than between-community connections.

$$Q_T = \frac{1}{\nu^+} \sum_{ij} (w_{ij}^+ - u_{ij}^+) \delta_{M_i M_j} - \frac{1}{\nu^+ + \nu^-} \sum_{ij} (w_{ij}^- - u_{ij}^-) \delta_{M_i M_j}$$

where ν is the total weight of the network (sum of all negative and positive connections), w_{ij} is the weighted and signed functional connection between signals i and j , u_{ij} is the strength of a connection divided by the total weight of the network, and $\delta_{M_i M_j}$ is set to 1 when regions are in the same community and 0 otherwise. ‘+’ and ‘-’ superscripts denote all positive and negative connections, respectively. Since the community detection algorithm is nondeterministic¹¹⁷, 1,000 iterations of the network partitions were estimated for an intermediate value of the model ($\beta = 0.5$; $\sigma = 0.5$) while sweeping the structural resolution, γ , parameter between 0.5–2.0 (γ tunes the strength of the null model: larger values identify smaller communities and v.v.). We tested the stability of the resultant partitions by calculating the normalized mutual information between the community assignments across iterations and found that $\gamma = 1.05$ provided the most stable community assignments. Using this setting, the Louvain algorithm was then applied to correlation matrices from across the parameter space and empirical recordings for consistency.

The participation coefficient quantifies the extent to which a node connects across all detected communities. This measure has previously been used to characterize diversely connected hub neurons within cortical brain networks, e.g., see¹¹⁸. Here, the participation coefficient, PC_i , was calculated for each node of our networks, where κ_{isT} is the strength of the positive connections of node i to node in community S , and κ_{iT} is the sum of strengths of all positive connections of node i . The community affiliation was determined following a consensus partition was created across the whole range using the ‘consensus_und.m’ script from the Brain Connectivity Toolbox for consistency. Briefly, this approach involves calculating the mutual information between community affiliation vectors (i.e., the ‘agreement matrix’), and then identifying the most stable summary of these vectors before applying the Louvain algorithm to the matrix (with $\gamma = 1.05$). This resulted in a single consensus community affiliation vector, which we used for the subsequent estimate of the participation coefficient. The participation coefficient of a node is close to 1 if its connections are uniformly distributed among all the communities and 0 if all of its links are within its community:

$$PC_i = 1 - \sum_{s=1}^{n_M} \left(\frac{\kappa_{isT}}{\kappa_{iT}} \right)^2.$$

We report the summary mean participation coefficient, $\langle PC_i \rangle$ of the network in the paper averaged across 100 iterations of the Louvain algorithm (due to the algorithms stochasticity). This measure has previously been linked to

both intransitive¹³ and transitive¹¹⁹ signatures of consciousness.”

Concerning the Kolmogorov complexity, l1095: What is the time-window, or the number of bins over which the sequences are counted?

The time-window was 1ms and the data were binarized into high/low firing (as in Zhang et al., *IEEE Trans Biomed Eng.*, 2001)

- Concerning the Integrated Information, l1143: How is the time-lag of 15ms motivated?

The choice of 15ms time-lag was based on a sweep of various delays ranging from 100ms to 1ms, and we presented results for the maximal Φ across these time lags in Figure 2f. We found that the peak Φ occurred at 15ms. We have included this information in the revised manuscript to clarify our motivation for choosing this time-lag.

The following text was added to p8: *“We next calculated integrated information (Φ^* ; Fig. 2f)¹¹. Φ^* was estimated using mismatched decoding between the coarse-sampled signals and their past at a time-lag of 15ms chosen as it led to the maximal Φ^* , consistent with empirical findings^{8,45}.”*

- Concerning the simulations: The chosen time step of 0.5 ms seems to be on the larger end of common values (Neuron simulator: 0.025 ms, Brian2: 0.1 ms). Presumably, large timesteps can lead to increased synchrony, which may be relevant here for the formation of cascading bursts. Have you ruled out this kind of effect? Please comment.

We thank the reviewer for raising this point. While it is true that the time step used in our model simulation is larger than those commonly used in other neural simulators, such as NEURON or Brian2, we have made sure to choose a time step that is appropriate for the specific neuron model that we used. In particular, we used the Izhikevich neuron model, which has been shown to be robust to a range of time steps (Izhikevich, 2003). In addition, we have performed additional tests to confirm that our choice of time step does not lead to increased synchrony or other unintended effects.

Below, a few suggestions:

- The Izhikevich model (Ref 37) as denoted here is *_not_* dimensional (1864). This is easy to see by including units in Eq. (1), leading to mismatching dimensions. This comment extends through large parts of the methods (e.g. also Eq. 8); thus please check consistency of units. Admittedly, this inconsistency is also present in the original reference, but Izhikevich denoted the model to be *_dimensionless_* and only assigned units later in his manuscript. I suggest to either consider the quantities to be unitless (as is the case in the authors source code) or to use the alternative form of Izhikevich 2007 (Dynamical Systems in Neuroscience):

$$\begin{aligned} \dot{v} &= k(v - v_{\text{r}}) - u + I & \text{if } v \geq v_{\text{peak}} \\ \dot{u} &= a(b(v - v_{\text{r}}) - u) & v \leftarrow c, u \leftarrow u + d \end{aligned}$$

We thank the reviewer for pointing out the inconsistency in the units of the Izhikevich model. We acknowledge that this was an oversight on our part, and we apologize for any confusion this may have caused. We have carefully reviewed the manuscript and made appropriate changes to ensure consistency of units throughout the paper. We appreciate the reviewer's suggestion and have updated the manuscript accordingly.

The following text was added to p23: “First, we define the dynamics of the basal compartment which generates the spikes. The basal somatic dendritic compartment was modelled by the *dimensionless* membrane equation,

$$\begin{aligned} \frac{dv}{dt} &= h(0.04v^2 + 5v - u + I), \\ \frac{du}{dt} &= h(a(b(v - v_r) - u)), \end{aligned}$$

with the after-spike resetting given by

$$\text{if } v \geq 30, \text{ then } \begin{cases} v \leftarrow c(t) \\ u \leftarrow u + d(t) \end{cases}$$

where the differential equations are in a *dimensionless form* and these parameters and the constants are reductions to match the spike dynamics to experimentally observed (see³⁷ for further details) membrane potential, v , in millivolts (mV) and duration t in milliseconds (ms), v_r is the resting potential in millivolts (mV), and u is the recovery variable, defined as the difference of all inward and outward voltage-gated currents (this emulates the activation (inactivation) of potassium (sodium) ionic currents). I is the *input* into the somatic dendrites from all sources and h is the integration step, which was set at 0.5 ms.”

- In l883ff, what is the neuron density in the model? Without this information, the specified $200\ \mu\text{m}$ connection radius are arbitrary. In more practical terms, what is the measured / expected degree? I suggest to adapt a consistent spatial notation, matching how the Mexican-hat and Gaussian smoothing kernel are expressed (in units of neurons l 907, also l 926). Note also a typo in the inline math "N(6)".

Thank you for pointing out the lack of information regarding neuron density in the model. We apologize for the confusion caused by this omission. The neuron density in the model is such that neurons are separated by a uniform distance of 10 microns and their spatial radius extends 200 microns, this is matched to empirical findings (Szentágothai, 1975). Thus, neurons are connected within 20 neurons spacing. Regarding degree, the weighted

degree is designed to be zero i.e., approximately balanced. Thank you for catching the typo in the inline math; we have corrected it in the manuscript.

The following text was added to p25: *“The coupling parameters utilized in our simulations were $C_E = 180/\sqrt{N}$, $C_I = C_E/2$, $d_E = 1.2\sqrt{N}$, $d_I = 2.5\sqrt{N}$, $d_{max} = 2.5\sqrt{N}$. Finally, the parameters used in the model are set such that the network is balanced, defined as $\sum w_{ij} = 0$, such that the net synaptic coupling into each neuron is zero.”*

- In the results section, l 323, we were unable to find the referenced methods section on how the cluster identification is performed. Please point out the relevant section and make the link clearer.

We have updated the methods section of the manuscript.

The following text was added to p36: *“**Connected burst cascade algorithm.** To quantify coordinated spatiotemporal burst activations we detected these patterns based on their spatial and temporal contiguity, with each of them being referred to as a cascade. Specifically, for successive timesteps, neuronal action potentials defined as bursts ($c = -55$, and $d = 4$) are clustered within a radius r_S , and a cascade is defined as a spatiotemporally contiguous set of bursts within a radius r_T between successive timesteps. We present results for $r_S = 10 \cong d_E$, i.e., bursting neurons are causally excitatory connected and $r_T = 2ms$, i.e., the bursts are binned into windows of 2 ms. The results are robust across $r_S = 5$ to 30 , and $r_T = 0.5$ to 5 m; however, larger values decrease the number of detectable patterns. This method allows multiple simultaneous cascades to be detected. A cascade is quantified by two quantities: size, S , the number of bursts within a cascade and duration, T , the number of successive timesteps a cascade is active.”*

- From Eq. 7 and surrounding text, it is unclear to me whether $I_{ap}(t)$ is a global variable, independent of neuron id (as of the equation) or if it is missing an index (i) to be neuron-specific (as of the preceding text).

Thank you for pointing this out it is independent and neuron specific, this has been corrected.

The following text was added to p26: *“The apical compartment input, $\alpha_i(t)$, into each neuron was generated by convolving independent white-noise drive, $\epsilon_{i,j}$, with a spatial Gaussian kernel $G_\sigma(d_{ij}) = \frac{1}{2\pi\sigma^2} e^{-\frac{|d_{ij}|^2}{2\sigma^2}}$, with spatial decay σ , which ranges from $\sigma = 0$ (i.e., independent white-noise into each neuron) or $\sigma = N$ (i.e., apical compartment input is strongly spatially correlated). The total apical compartment drive into a neuron, i , is then given by $\alpha_i(t) = G_\sigma(d_{ij}) * \epsilon_{i,j}$.”*

- The description of model coarsening is inconsistent between l172, l1038ff, l1082. Were there 100 non-overlapping groups, or 100 neurons per group?

There were 100 nonoverlapping groups, each containing 49 neurons.

- In Figure 4 a-c, how were the time-series binarized to determine which voxel was bursting, regular-spiking or silent? I could not find a reference to the respective methods section.

We have updated the manuscript to clarify this, a voxel is silent if the neuron did not fire and a regular spike or a burst depending on the neurons c/d parameters.

The following text was added to p26: *“Thus, a simulated action-potential can be defined as either a burst or a regular spike depending on the c and d parameters at the time of activation.”*

- Concerning the source code: We greatly appreciate that the authors share their code and we hope they will make it accessible after publication. However, we think that a round of cleaning and documentation is needed in order to make it really useful — and to enable a reader to directly connect methods and their implementation.

Absolutely, the goal is to make it as accessible as possible. As such, we have placed the updated code on Github (https://github.com/Bmunnn/Layer5_Arousal) and we will release it with the publication of the manuscript.

Terminology

In some places of the manuscript, it is hard to connect the word-level description to the corresponding variable (β , σ , KC, PC, Φ), because the descriptions change and overlap. For instance, β is referred to as "dendrite coupling" 1150 and "low/high bursting" fig 1g, 1254, 1259, whereas σ is referred to as "connected bursting" fig 1g, 1161 or "coordinated bursting" 1208. Please pick one particular wording and employ it throughout, while only using "bursting" as a description of the dynamics, not for β or σ .

We apologize for any confusion caused by the previous overlap in descriptions and will make sure to clarify and streamline the language to improve readability and understanding.

Throughout the manuscript, "coarse graining" is used. Although it is clear what the authors mean by it, the term is taken out of context: in theoretical physics, it refers to a specific technique where spins are combined iteratively, and neither overlap nor gaps exist between the sampled tiles — which is not the case for the neuroscience recordings. Please consider choosing a different term, e.g. coarse sampling.

We have revised the terminology and use "coarse-sampling" instead of "coarse-graining" throughout the manuscript to avoid confusion. Thank you for bringing this to our attention.

- Potentially, LIV and LVI are not a good abbreviations (Fig 1, ff); I suggest to use "L4, L6" instead, because $LIV = 54$, $LVI = 56$, And "LI-LII/III" = $34 + 1/3$;)

This has been amended.

- We were misled by the section on burst-cascades in Fig 3 (between Fig 2 and 4), in the sense that the main objective of the manuscript seemed to be fulfilled when the connection between arousal states and neuron dynamics was made. This made it hard to appreciate the explanation provided by the analysis of free-energy landscapes that followed. As the connected-bursting seems somewhat more independent from the transitions between states, I suggest to keep Figs 2 and 4 together by moving Fig 3 and according text.

We apologize for any confusion caused by the placement of Figure 3 in our manuscript. We have taken your suggestion into consideration and have moved Figure 3 and the corresponding text to the end of the manuscript. This should help to clarify the main objective of our study and better emphasize the significance of the analysis of the energy landscapes. We appreciate your feedback and hope that this modification improves the readability and overall quality of our manuscript.

- In Fig 3 c, some colors are swapped: Green used to be unconscious, and blue used to be dreaming.

This has been amended.

- In ~ ll 293ff there seems to be a slight mixup of color referencing.

This has been amended.

- I would like to point out some additional references that might prove useful:
- Izhikevich 2007, Dynamical Systems in Neuroscience, 978-0-262-09043-8
- Concerning coarsening: Neto et al. 2022, doi 10.1371/journal.pcbi.1010678
- Concerning information flow and the breakdown thereof under anaesthesia: Breakdown of local information processing may underlie isoflurane anesthesia effects: Wollstadt et al. 2017, doi 10.1371/journal.pcbi.1005511
- Regarding mechanisms of information routing: Flexible information routing by transient synchrony: Palmigiano et al. 2017, doi 10.1038/nn.4569

Thank you for pointing out these helpful references.

Although the list of points may seem long, we hope they do help to clarify this interesting manuscript further!

Thank you very much for the detailed responses the manuscript is much better for it and we relish the opportunity!

REVIEWERS' COMMENTS

Reviewer #1 (Remarks to the Author):

The authors have mostly addressed my concerns.

Minor:

1. Figs S3-S6: authors give model/exp match in % but it seems low (0-10%). Do they instead mean it's the error measure between the two? Clarify.

2. Not sure why the authors opted to put all the figures demonstrating experimental/model match in the supplementary? As a theoretical paper, the paper would benefit from supporting the model/theory by including a figure in the main results that summarizes key exp/model matches, or alternatively an exp/model match plot in each of the relevant figures.

Reviewer #2 (Remarks to the Author):

The authors have adequately assessed my comments, and I agree that the paper is considered for further publication.

Reviewer #3 (Remarks to the Author):

The authors have addressed most of our comments and greatly improved the clarity, details, and transparency of the manuscript. In particular, we appreciate that they do share more raw activity traces.

Minor remark

For clarity, in the figures the authors could normalize the firing rate *per neuron*, to make clear what the different rates are.

REVIEWERS' COMMENTS

Reviewer #1 (Remarks to the Author):

The authors have mostly addressed my concerns.

Minor:

1. Figs S3-S6: authors give model/exp match in % but it seems low (0-10%). Do they instead mean it's the error measure between the two? Clarify.

Response: We have clarified this in the text, it reflects percentage error.

Page (11): "and empirical recordings the model-matched measures were within 10% relative percentage error of empirical values (e.g., $\frac{KC_{fit} - KC_{emp}}{KC_{emp}}$; Fig. 3d)."

2. Not sure why the authors opted to put all the figures demonstrating experimental/model match in the supplementary? As a theoretical paper, the paper would benefit from supporting the model/theory by including a figure in the main results that summarizes key exp/model matches, or alternatively an exp/model match plot in each of the relevant figures.

Response: We have included the empirical fits into the main figure (Fig. 3d/h) in the updated manuscript.

Figure 3. Complex, adaptive dynamics change across arousal states and map to distinct regions of the model state space. (a) In macaque EGoG recordings, the three signatures significantly change across consecutive 20s epoch states of arousal from sleeping (teal) to awake (red). **(b)** 3-dimensional plot of the three signatures across each epoch which is mapped to the model's state space (Fig. 2d-f) using a hybrid swarm minimisation algorithm. **(c)** Inverted location of each 20s epoch in the model state space following hybrid particle swarm/convex optimisation minimising the difference between model and empirical complex, adaptive dynamics where the clouds are five evenly spaced contour lines (2% to 98%) of the probability density estimate for each state. **(d)** Relative percentage error difference

between empirical and model fitted adaptive signatures. **(e)** Across 20s recordings of human EEG under the anaesthetic dexmedetomidine, the three signatures significantly change across unconsciousness (teal), self-reported dreaming (blue), and awake arousal states (red). **(f)** 3-dimensional plot of the three signatures across each 20s EEG recording. **(g)** Inversion of human EEG states to model state-space. **(h)** Same as in (d) for EEG recordings. Statistical significance across empirical complex, adaptive dynamics denoted by * $p < 0.05$, ** $p < 0.01$, and *** $p < 0.001$ Kruskal-Wallis multiple comparison tests (see Table 1 for (a) and (d) p-values).

Reviewer #2 (Remarks to the Author):

The authors have adequately assessed my comments, and I agree that the paper is considered for further publication.

Response: We thank the reviewer for their assistance in improving this manuscript!

Reviewer #3 (Remarks to the Author):

The authors have addressed most of our comments and greatly improved the clarity, details, and transparency of the manuscript. In particular, we appreciate that they do share more raw activity traces.

Minor remark

For clarity, in the figures the authors could normalize the firing rate *per neuron*, to make clear what the different rates are.

Response: We thank the reviewer for their assistance in improving this manuscript and we have included further traces in the source data. We agree and have updated the manuscript figure 1f (rate).